# Adaptive Sampling for Stochastic Risk-Averse Learning

**Sebastian Curi**
Dept. of Computer Science
ETH Zurich
scuri@inf.ethz.ch

**Kfir Y. Levy**
Faculty of Electrical Engineering
Technion
kfirylevy@technion.ac.il

**Stefanie Jegelka**
CSAIL
MIT
stefje@mit.edu

**Andreas Krause**
Dept. of Computer Science
ETH Zurich
krausea@inf.ethz.ch

## Abstract

In high-stakes machine learning applications, it is crucial to not only perform well *on average*, but also when restricted to *difficult* examples. To address this, we consider the problem of training models in a risk-averse manner. We propose an adaptive sampling algorithm for stochastically optimizing the *Conditional Value-at-Risk (CVaR)* of a loss distribution, which measures its performance on the $\alpha$ fraction of most difficult examples. We use a distributionally robust formulation of the CVaR to phrase the problem as a zero-sum game between two players, and solve it efficiently using regret minimization. Our approach relies on sampling from structured Determinantal Point Processes (DPPs), which enables scaling it to large data sets. Finally, we empirically demonstrate its effectiveness on large-scale convex and non-convex learning tasks.

## 1  Introduction

Machine learning systems are increasingly deployed in high-stakes applications. This imposes reliability requirements that are in stark discrepancy with how we currently train and evaluate these systems. Usually, we optimize *expected performance* both in training and evaluation via empirical risk minimization (Vapnik, 1992). Thus, we sacrifice occasional large losses on "difficult" examples in order to perform well on average. In this work, we instead consider a *risk-averse* optimization criterion, namely the *Conditional Value-at-Risk* (CVaR), also known as the Expected Shortfall. In short, the $\alpha$-CVaR of a loss distribution is the average of the losses in the $\alpha$-tail of the distribution.

Optimizing the CVaR is well-understood in the *convex* setting, where duality enables a reduction to standard empirical risk minimization using a modified, truncated loss function from Rockafellar et al. (2000). Unfortunately, this approach fails when *stochastically* optimizing the CVaR – especially on non-convex problems, such as training deep neural network models. A likely reason for this failure is that mini-batch estimates of gradients of the CVaR suffer from high variance.

To address this issue, we propose a novel *adaptive sampling algorithm* – ADA-CVAR (Section 4). Our algorithm initially optimizes the mean of the losses but gradually adjusts its sampling distribution to increasingly sample tail events (difficult examples), until it eventually minimizes the CVaR (Section 4.1). Our approach naturally enables the use of standard stochastic optimizers (Section 4.2). We provide convergence guarantees of the algorithm (Section 4.3) and an efficient implementation (Appendix A). Finally, we demonstrate the performance of our algorithm in a suite of experiments (Section 5).

## 2   Related Work

**Risk Measures**   Risk aversion is a well-studied human behavior, in which agents assign more weight to adverse events than to positive ones (Pratt, 1978). Approaches for modeling risk include using utility functions that emphasize larger losses (Rabin, 2013); prospect theory that re-scales the probability of events (Kahneman and Tversky, 2013); or direct optimization of coherent risk-measures (Artzner et al., 1999). Rockafellar et al. (2000) introduce the CVaR as a particular instance of the latter class. The CVaR has found many applications, such as portfolio optimization (Krokhmal et al., 2002) or supply chain management (Carneiro et al., 2010), as it does not rely on specific utility or weighing functions, which offers great flexibility.

**CVaR in Machine Learning**   The $\nu$-SVM algorithm by Schölkopf et al. (2000) can be interpreted as optimizing the CVaR of the loss, as shown by Gotoh and Takeda (2016). Also related, Shalev-Shwartz and Wexler (2016) propose to minimize the *maximal loss* among all samples. The maximal loss is the limiting case of the CVaR when $\alpha \to 0$. Fan et al. (2017) generalize this work to the top-$k$ average loss. Although they do not mention the relationship to the CVaR, their learning criterion is equal to the CVaR for empirical measures. For optimization, they use an algorithm proposed by Ogryczak and Tamir (2003) to optimize the maximum of the sum of $k$ functions; this algorithm is the same as the "truncated" algorithm of Rockafellar et al. (2000) to optimize the CVaR. Recent applications of the CVaR in ML include risk-averse bandits (Sani et al., 2012), risk-averse reinforcement learning (Chow et al., 2017), and fairness (Williamson and Menon, 2019). All these use the original "truncated" formulation of Rockafellar et al. (2000) to optimize the CVaR. One of the major shortcomings of this formulation is that mini-batch gradient estimates have high variance. In this work, we address this via a method based on adaptive sampling, inspired by Shalev-Shwartz and Wexler (2016), that allows us to handle large datasets and complex (deep neural network) models.

**Distributionally Robust Optimization**   The CVaR also has a natural *distributionally robust optimization* (DRO) interpretation (Shapiro et al., 2009, Section 6.3), which we exploit in this paper. For example, Ahmadi-Javid (2012) introduces the entropic value-at-risk by considering a different DRO set. Duchi et al. (2016); Namkoong and Duchi (2017); Esfahani and Kuhn (2018); Kirschner et al. (2020) address related DRO problems, but with different uncertainty sets. We use the DRO formulation of the CVaR to phrase its optimization as a game. To solve the game, we propose an adaptive algorithm for the learning problem. Our algorithm is most related to (Namkoong and Duchi, 2016), who develop an algorithm for DRO sets induced by Cressie-Read $f$-divergences. Instead, we use a different DRO set that arises in common data sets (Mehrabi et al., 2019) and we provide an *efficient* algorithm to solve the DRO problem in large-scale datasets.

## 3   Problem Statement

We consider supervised learning with a *risk-averse learner*. The learner has a data set comprised of i.i.d. samples from an unknown distribution, i.e., $D = \{(x_1, y_1), \dots (x_N, y_N)\} \in (\mathcal{X} \times \mathcal{Y})^N \sim \mathcal{D}^N$, and her goal is to learn a function $h_\theta : \mathcal{X} \to \mathcal{R}$ that is parametrized by $\theta \in \Theta \subset \mathbb{R}^d$. The performance of $h_\theta$ at a data point is measured by a *loss function* $l : \Theta \times \mathcal{X} \times \mathcal{Y} \to [0, 1]$. We write the random variable $L_i(\theta) = l(\theta; x_i, y_i)$. The learner's goal is to minimize the *CVaR of the losses* on the (unknown) distribution $\mathcal{D}$ w.r.t. the parameters $\theta$.

**CVaR properties**   The CVaR of a random variable $L \sim P$ is defined as $\mathbb{C}^\alpha[L] = \mathbb{E}_P[L|L \geq \ell^\alpha]$, where $\ell^\alpha$ is the $1 - \alpha$ quantile of the distribution, also called the *Value-at-Risk* (VaR). We illustrate the mean, VaR and CVaR of a typical loss distribution in Figure 1. It can be shown that the CVaR of a random variable has a natural *distributionally robust optimization* (DRO) formulation, namely as the expected value of the same random variable under a *different* law. This law arises from the following optimization problem (Shapiro et al., 2009, Sec. 6.3):

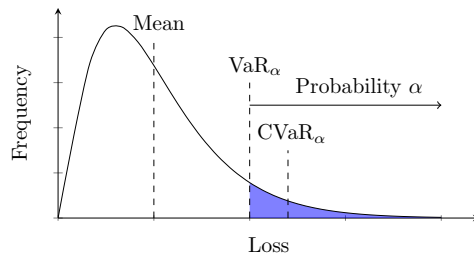

Figure 1: CVaR of a loss distribution

$$\mathbb{C}^\alpha[L] = \max_{Q \in \mathcal{Q}^\alpha} \mathbb{E}_Q[L], \qquad (1)$$

where $\mathcal{Q}^\alpha = \left\{ Q \ll P, \frac{dQ}{dP} \leq \frac{1}{\alpha} \right\}$. Here, $Q \ll P$ means that $Q$ is absolutely continuous w.r.t. $P$. The distribution $Q^\star$ that solves Problem (1) places all the mass uniformly in the tail, i.e., the blue shaded region of Figure 1. Thus, optimizing the CVaR can be viewed as *guarding against a particular kind of distribution shift*, which reweighs arbitrary parts of the data up to a certain amount $\frac{1}{\alpha}$. Rockafellar et al. (2000) prove strong duality for Problem (1). The dual objective is:

$$\mathbb{C}^\alpha[L] = \min_{\ell \in \mathbb{R}} \ell + \frac{1}{\alpha} \mathbb{E}_P \left[ \max \{0, L - \ell\} \right]. \qquad (2)$$

**Learning with the CVaR**  Problem (2) can be used to estimate the CVaR of a random variable by replacing the expectation $\mathbb{E}_P$ by the empirical mean $\hat{\mathbb{E}}$, yielding

$$\min_{\ell \in \mathbb{R}, \theta \in \Theta} \ell + \frac{1}{\alpha N} \sum_{i=1}^N \left[ \max \{0, L_i(\theta) - \ell\} \right]. \qquad (3)$$

For convex $L_i$, Problem (3) has computable subgradients, and hence lends itself to subgradient-based optimization. Furthermore, in this case Problem (3) is jointly convex in $(\ell, \theta)$. We refer to this standard approach as TRUNC-CVAR, as it effectively optimizes a modified loss, truncated at $\ell$.

Problem (3) is indeed a sensible learning objective in the sense that the empirical CVaR concentrates around the population CVaR uniformly for all functions $h \in \mathcal{H}$.

**Proposition 1.** *Let $h : \mathcal{X} \to \mathcal{Y}$ be a finite function class $|\mathcal{H}|$. Let $L(h) : \mathcal{H} \to [0, 1]$ be a random variable. Then, for any $0 < \alpha \leq 1$, with probability at least $1 - \delta$,*

$$\mathbb{E} \left[ \sup_{h \in \mathcal{H}} \left| \widehat{\mathbb{C}}^\alpha[L(h)] - \mathbb{C}^\alpha[L(h)] \right| \right] \leq \frac{1}{\alpha} \sqrt{\frac{\log(2|\mathcal{H}|/\delta)}{N}}.$$

*Proof.* See Appendix B.1. □

The result above is easily extended to classes $\mathcal{H}$ with finite VC (pseudo-)dimension. Concurrent to this work, Lee et al. (2020), Soma and Yoshida (2020), and Mhammedi et al. (2020) present similar statistical rates based on different assumptions.

**Challenges for Stochastic Optimization**  In the common case that a variant of SGD is used to optimize the learning problem (3), the expectation is approximated with a mini-batch of data. But, when this batch is sampled uniformly at random from the data, only a fraction $\alpha$ of points will contain gradient information. The gradient of the remaining points gets truncated to zero by the $\max\{\cdot\}$ non-linearity. Furthermore, the gradient of the examples that *do* contain information is scaled by $1/\alpha$, leading to exploding gradients. These facts make stochastic optimization of Problem (3) extremely challenging, as we demonstrate in Section 5.

Our key observation is that the root of the problem lies in the *mismatch* between the sampling distribution $P$ and the unknown distribution $Q^\star$, from which we would ideally like to sample. In fact, Problem (3) can be interpreted as a form of rejection sampling – samples with losses smaller than $\ell$ are rejected. It is well known that Monte Carlo estimation of rare events suffers from high variance (Rubino and Tuffin, 2009). To address this issue, we propose a novel sampling algorithm that *adaptively* learns to sample events from the distribution $Q^\star$ while optimizing the model parameters $\theta$.

## 4  ADA-CVAR: Adaptive Sampling for CVaR Optimization

We directly address the DRO problem (1) on the empirical measure $\hat{P}$ for learning. The DRO set is $\mathcal{Q}^\alpha = \left\{ q \in \mathbb{R}^N \mid 0 \leq q_i \leq \frac{1}{k}, \sum_i q_i = 1 \right\}$ with $k = \lfloor \alpha N \rfloor$. The learning problem becomes:

$$\min_{\theta \in \Theta} \max_{q \in \mathcal{Q}^\alpha} \mathbb{E}_q[L_i(\theta)] = \min_{\theta \in \Theta} \max_{q \in \mathcal{Q}^\alpha} q^\top L(\theta), \qquad (4)$$

where $L(\theta) \in \mathbb{R}^N$ has $i$-th index $L_i(\theta)$. The learning problem (4) is a minimax game between a $\theta$-player (the learner), whose goal is to *minimize* the objective function by selecting $\theta \in \Theta$, against a $q$-player (the sampler), whose goal is to *maximize* the objective function by selecting $q \in \mathcal{Q}^\alpha$.

To solve the game (4), we use techniques from regret minimization with partial (bandit) feedback. In particular, we exploit that one can solve minimax games by viewing both players as online learners

that compete, and by equipping each of them with no-regret algorithms (Freund and Schapire, 1999). With the partial (bandit) feedback model, we only need to consider a *small* subset of the data in each iteration of the optimization algorithm. In contrast, full-information feedback would require a full pass over the data per iteration, invalidating all benefits of stochastic optimization.

Next, we describe and analyze an online learning algorithm for each of the two players and prove guarantees with respect to the DRO problem (4). We outline the final algorithm, which we call ADA-CVAR, in Algorithm 1, where we use an adaptive sampling scheme for the $q$-player and SGD for the $\theta$-player. Initially, the $q$-player (sampler) plays the uniform distribution and the $\theta$-player (learner) selects any parameter in the set $\Theta$. In iteration $t$, the sampler samples a data point (or a mini-batch) with respect to the distribution $q_t$. Then, the learner performs an SGD step on the sample(s) selected by the sampler player[1]. Finally, the $q$-player adapts the distribution to favor examples with higher loss and thus maximize the objective in (4).

---

**Algorithm 1:** ADA-CVAR

**input** Learning rates $\eta_s, \eta_l$.
1: **Sampler:** Initialize k-DPP $w_1 = \mathbf{1}_N$.
2: **Learner:** Initialize parameters $\theta_0 \in \Theta$.
3: **for** $t = 1, \dots, T$ **do**
4:   **Sampler:** Sample data point
      $i_t \sim q_t = \frac{1}{k}\mathbb{P}_{w_t}(i)$.
5:   **Learner:** $\theta_t = \theta_{t-1} - \eta_l \nabla L_{i_t}(\theta_{t-1})$.
6:   **Sampler**: Build estimate
      $\hat{L}_t = \frac{L_{i_t}(\theta_t)}{q_{t,i_t}}[[i == i_t]]$.
7:   **Sampler**: Update k-DPP
      $w_{t+1,i} = w_{t,i}e^{\eta_s \hat{L}_{t,i}}$.
8: **end for**
**output** $\bar{\theta}, \bar{q} \sim_{u.a.r} \{(\theta_t, q_t)\}_{t=1}^T$

---

### 4.1 Sampler ($q$-Player) Algorithm

In every iteration $t$, the learner player sets a vector of losses through $\theta_t$. We denote by $L(\theta_t; x_i, y_i) = L_{t,i}$ the loss at time $t$ for example $i$ and by $L(\theta_t) = L_t$ the vector of losses. The sampler player chooses an index $i_t$ (or a mini-batch) and a vector $q_t$. Then, only $L_{t,i_t}$ is revealed and she suffers a cost $q_t^\top L_t$. In such setting, the best the player can aim to do is to minimize its *regret*:

$$\text{SR}_T := \max_{q \in \mathcal{Q}^\alpha} \sum_{t=1}^{T} q^\top L_t - \sum_{t=1}^{T} q_t^\top L_t. \tag{5}$$

The regret measures how good the sequence of actions of the sampler is, compared to the best single action (i.e., distribution over the data) in hindsight, after seeing the sequence of iterates $L_t$. The sampler player problem is a linear adversarial bandit. Exploration and sampling in this setting are hard (Bubeck et al., 2012). Our efficient implementation exploits the specific combinatorial structure.

In particular, the DRO set $\mathcal{Q}^\alpha$ is a polytope with $\binom{N}{k}$ vertices, each corresponding to a different subset $I$ of size $k$ of the ground set $2^{[N]}$. As the inner optimization problem over $q$ in (4) is a linear program, the optimal solution $q^\star$ is a vertex. Thus, the sampler problem can be reduced to a *best subset selection* problem: find the best set among all size-$k$ subsets $\mathcal{I}_k = \{I \subseteq 2^{[N]} \mid |I| = k\}$. Here, the value of a set $I$ at time $t$ is simply the average of the losses $(1/k)\sum_{i \in I} L_i(\theta_t)$. The problem of maximizing the value over time $t$ can be viewed as a *combinatorial bandit* problem, as we have a combinatorial set of "arms", one per $I \in \mathcal{I}_k$ (Lattimore and Szepesvári, 2018, Chapter 30). Building on Alatur et al. (2020), we develop an efficient algorithm for the sampler.

**Starting Point: EXP.3** A well known no-regret bandit algorithm is the celebrated EXP.3 algorithm (Auer et al., 2002). Let $\Delta_I := \left\{ \tilde{W} \in \mathbb{R}^{\binom{N}{k}} \mid \sum_I \tilde{W}_I = 1, \tilde{W}_I \geq 0 \right\}$ be the simplex of distributions over the $\binom{N}{k}$ subsets. Finding the best distribution $W_I^\star \in \Delta_I$ is equivalent to finding the best subset $I^\star \in \mathcal{I}_k$. By transitivity, this is equivalent to finding the best $q^\star \in \mathcal{Q}^\alpha$. To do this, EXP.3 maintains a vector $W_{I,t} \in \Delta_I$, samples an element $I_t \sim W_{I,t}$ and observes a loss associated with element $I_t$. Finally, it updates the distribution using multiplicative weights. Unfortunately, EXP.3 is *intractable in two ways*: Sampling a $k$-subset $I_t$ would require evaluating the losses of $k = \lfloor \alpha N \rfloor$ data points, which is impractical. Furthermore, the naive EXP.3 algorithm is intractable because the dimension of $W_{I,t}$ is *exponential* in $k$. In turn, the regret of this algorithm also depends on the dimension of $W_{I,t}$.

**Efficiency through Structure** The crucial insight is that we can *exploit the combinatorial structure of the problem and additivity of the loss* to exponentially improve efficiency. First, we exploit that

weights of individual elements and sets of them are related by $W_{t,I} = \sum_{i \in I} w_{t,i}$. Thus, instead of observing the loss $L_{I_t}$, we let the $q$-player sample only a *single element* $i_t$ uniformly at random from the set $I_t \sim W_{I,t}$, observe its loss $L_{i_t}$, and use it to update a weight vector $w_{t,i}$. The single element $i_t$ sampled by the algorithm provides information about the loss of all $\binom{N-1}{k-1}$ *sets* that contain $i_t$. This allows us to obtain regret guarantees that are sub-linear in $N$ (rather than in $N^k$). Second, we *exponentially improve computational cost* by developing an algorithm that maintains a vector $w \in \mathbb{R}^N$ and uses *k-Determinantal Point Processes* to map it to distributions over subsets of size $k$.

**Definition 4.1** (k-DPP, Kulesza et al. (2012)). A $k$-Determinantal Point Process over a ground set $N$ is a distribution over all subsets of size $k$ s.t. the probability of a set is $\mathbb{P}(I) \propto \det(K_I)$, where $K$ is a positive definite kernel matrix and $K_I$ is the submatrix of $K$ indexed by $I$. ∎

In particular, we consider k-DPPs with *diagonal* kernel matrices $K = \operatorname{diag} w$, with $w \in \mathbb{R}^N_{\geq 0}$ and at least $k$ strictly positive elements. This family of distributions is sufficient to contain, for example, the uniform distribution over the $\binom{N}{k}$ subsets and all the vertices of $\mathcal{Q}^\alpha$. We use such k-DPPs to *efficiently* map a vector of size $N$ to a distribution over $\binom{N}{k}$ subsets. We also denote the marginal probability of element $i$ by $\mathbb{P}_w(i)$. It is easy to verify that the vector of marginals $\frac{1}{k}\mathbb{P}_w(i) \in \mathcal{Q}^\alpha$. Hence, we directly use the k-DPP marginals as the sampler's decision variables.

We can finally describe the sampler algorithm. We initialize the k-DPP kernel with the uniform distribution $w_1 = \mathbf{1}_N$. In iteration $t$, the sampler plays the distribution $q_t = \frac{1}{k}\mathbb{P}_{w_t}(\cdot) \in \mathcal{Q}^\alpha$ and samples an element $i_t \sim q_t$. The loss at index $i_t$, $L_{t,i_t}$, is revealed to the sampler and only the index $i_t$ of $w_t$ is updated according to the multiplicative update $w_{t+1,i_t} = w_{t+1,i_t} e^{kL_{t,i_t}/q_{t,i_t}}$.

This approach addresses the disadvantages of the EXP.3 algorithm. Computationally, it only requires $O(N)$ memory. After sampling every element $i_t$, the distribution over the $\binom{N-1}{k-1}$ sets that contain $i_t$ are updated. This yield rates that depend sub-linearly on the data set size which we prove next.

**Lemma 1.** *Let the sampler player play the* ADA-CVAR *Algorithm with* $\eta_s = \sqrt{\frac{\log N}{NT}}$. *Then, for any sequence of losses she suffers a regret* (5) *of at most* $O(\sqrt{TN \log N})$.

*Proof sketch.* For a detailed proof please refer to Appendix B.2. First, we prove in Proposition 3 that the iterates of the algorithm are effectively in $\mathcal{Q}^\alpha$. Next, we prove in Proposition 4 that the comparator in the regret of Alatur et al. (2020) and in the sampler regret (5) have the same value (scaled by $k$). Finally, the result follows as a corollary from these propositions and Alatur et al. (2020, Lemma 1). □

### 4.2 Learner ($\theta$-Player) Algorithm

Analogous to the sampler player, the learner player seeks to minimize its regret

$$\text{LR}_T := \sum_{t=1}^T q_t^\top L(\theta_t) - \min_{\theta \in \Theta} \sum_{t=1}^T q_t^\top L(\theta). \tag{6}$$

Crucially, the learner can choose $\theta_t$ *after* the sampler selects $q_t$. Thus, the learner can play the Be-The-Leader (BTL) algorithm:

$$\theta_t = \arg\min_{\theta \in \Theta} \sum_{\tau=1}^t q_\tau^\top L(\theta) = \arg\min_{\theta \in \Theta} \bar{q}_t^\top L(\theta), \tag{7}$$

where $\bar{q}_t = \frac{1}{t}\sum_{\tau=1}^t q_\tau$ is the average distribution (up to time $t$) that the sampler player proposes.

Instead of assuming access to an exact optimizer, we assume to have an ERM oracle available.

**Assumption 1** ($\epsilon_{\text{oracle}}$-correct ERM Oracle). *The learner has access to an ERM oracle that takes a distribution $q$ over the dataset as input and outputs $\hat{\theta}$, such that*

$$q^\top L(\hat{\theta}) \leq \min_{\theta \in \Theta} q^\top L(\theta) + \epsilon_{\text{oracle}}.$$

To implement the ERM Oracle, the learner player must solve a *weighted* empirical loss minimization in Problem (7) in every round. For non-convex problems, this is in general NP-hard (Murty and Kabadi, 1987), so obtaining efficient and provably no-regret guarantees in the non-convex setting seems unrealistic in general.

Despite this hardness, the success of deep learning empirically demonstrates that stochastic optimization algorithms such as SGD are able to find very good (even if not necessarily optimal) solutions for the ERM-like non-convex problems. Furthermore, SGD on the sequence of samples $\{i_\tau \sim q_\tau\}_{\tau=1}^t$ approximately solves the BTL problem. To see why, we note that such sequence of samples is an unbiased estimator of $\bar{q}_t$ from the BTL algorithm (7). Then, for the freshly sampled $i_t \sim q_t$, a learner that chooses $\theta_t := \theta_{t-1} - \eta_l \nabla L_{i_t}(\theta_{t-1})$ is (approximately) solving the BTL algorithm with SGD.

**Lemma 2.** *A learner player that plays the BTL algorithm with access to an ERM oracle as in Assumption 1, achieves at most $\epsilon_{\mathrm{oracle}} T$ regret.*

*Proof.* See Appendix B.3. □

For convex problems, we know that it is not necessary to solve the BTL problem (7), and algorithms such as online projected gradient descent (Zinkevich, 2003) achieve no-regret guarantees. As shown in Appendix C, the learner suffers $\mathrm{LR}_T = O(\sqrt{T})$ regret by playing SGD in convex problems.

### 4.3 Guarantees for CVaR Optimization

Next, we show that if both players play the no-regret algorithms discussed above, they solve the game (4). Using $J(\theta, q) := q^\top L(\theta)$, the minimax equilibrium of the game is the point $(\theta^\star, q^\star)$ such that $\forall \theta \in \Theta, q \in \mathcal{Q}^\alpha; \ J(\theta^\star, q) \le J(\theta^\star, q^\star) \le J(\theta, q^\star)$. We assume that this point exists (which is guaranteed, e.g., when the sets $\mathcal{Q}^\alpha$ and $\Theta$ are compact). The game regret is

$$\mathrm{GameRegret}_T := \sum_{t=1}^{T} J(\theta_t, q^\star) - J(\theta^\star, q_t). \tag{8}$$

**Theorem 1** (Game Regret). *Let $L_i(\cdot) : \Theta \to [0, 1]$, $i = \{1, ..., N\}$ be a fixed set of loss functions. If the sampler plays* ADA-CVAR *and the learner plays the oracle-BTL algorithm, then the game has regret $O(\sqrt{TN \log N} + \epsilon_{\mathrm{oracle}} T)$.*

*Proof sketch.* We bound the Game regret with the sum of the learner and sampler regret and use the results of Lemma 2 and Lemma 1. For a detailed proof please refer to Appendix B.4. □

This immediately implies our main theoretical result, namely a performance guarantee for the solution obtained by ADA-CVAR for the central problem of minimizing the empirical CVaR (4).

**Corollary 1** (Online to Batch Conversion). *Let $L_i(\cdot) : \Theta \to [0, 1]$, $i = \{1, ..., N\}$ be a set of loss functions sampled from a distribution $\mathcal{D}$. Let $\theta^\star$ be the minimizer of the CVaR of the empirical distribution $\hat{\mathbb{C}}^\alpha$. Let $\bar{\theta}$ be the output of* ADA-CVAR, *selected uniformly at random from the sequence $\{\theta_t\}_{t=1}^T$. Its expected excess CVaR is bounded as:*

$$\mathbb{E}\hat{\mathbb{C}}^\alpha[L(\bar{\theta})] \le \hat{\mathbb{C}}^\alpha[L(\theta^\star)] + O(\sqrt{N \log N / T}) + \epsilon_{\mathrm{oracle}}$$

*where the expectation is taken w.r.t. the randomization in the algorithm, both for the sampling steps and the randomization in choosing $\bar{\theta}$.*

*Proof sketch.* For a detailed proof please refer to Appendix B.5. The excess CVaR is bounded by the duality gap, which in turn is upper-bounded by the average game regret. □

**Corollary 2** (Population Guarantee). *Let $L(\cdot) : \Theta \to [0, 1]$ be a Random Variable induced by the data distribution $\mathcal{D}$. Let $\theta^\star$ be the minimizer of the CVaR at level $\alpha$ of such Random Variable. Let $\bar{\theta}$ be the output of* ADA-CVAR, *selected uniformly at random from the sequence $\{\theta_t\}_{t=1}^T$. Then, with probability at least $\delta$ the expected excess CVaR of $\bar{\theta}$ is bounded as:*

$$\mathbb{E}\mathbb{C}^\alpha[L(\bar{\theta})] \le \mathbb{C}^\alpha[L(\theta^\star)] + O(\sqrt{N \log N / T}) + \epsilon_{\mathrm{oracle}} + \epsilon_{\mathrm{stat}}$$

*where $\epsilon_{\mathrm{stat}} = \tilde{O}(\frac{1}{\alpha}\sqrt{\frac{1}{N}})$ comes from the statistical error and the expectation is taken w.r.t. the randomization in the algorithm, both for the sampling steps and the randomization in choosing $\bar{\theta}$.*

*Proof sketch.* For a detailed proof please refer to Appendix B.6. We bound the statistical error using Proposition 1 and the optimization error is bounded using Corollary 1. □

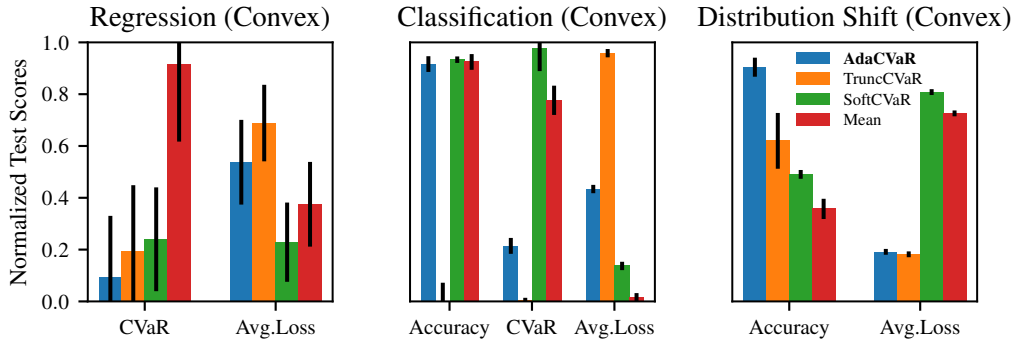

Figure 2: Scores are normalized between 0 and 1 to compare different data sets. *Left*: Linear regression tasks. ADA-CVAR has lower CVaR than benchmarks. *Middle*: Binary classification (logistic regression) tasks. ADA-CVAR obtains the same accuracy as MEAN and SOFT-CVAR with lower CVaR. TRUNC-CVAR outputs an approximately uniform distribution yielding low CVaR but poor predictive accuracy. *Right*: Binary classification (logistic regression) tasks with train/test 90% distribution shift. ADA-CVAR has the highest test accuracy and low average surrogate loss.

It is instructive to consider the special cases $k = 1$ and $k = N$. For $k = N$, $q_t$ remains uniform and ADA-CVAR reduces to SGD. For $k = 1$, the sampler simply plays standard EXP.3 over data points and ADA-CVAR reduces to the algorithm of Shalev-Shwartz and Wexler (2016) for the max loss.

## 5   Experiments

In our experimental evaluation, we compare ADA-CVAR on both convex (linear regression and classification) and non-convex (deep learning) tasks. In addition to studying how it performs in terms of the CVaR and empirical risk on the training and test set, we also investigate to what extent it can help guard against distribution shifts. In Appendix D, we detail the experimental setup. We provide an open-source implementation of our method, which is available at `http://github.com/sebascuri/adacvar`.

**Baseline Algorithms**   We compare our adaptive sampling algorithm (ADA-CVAR) to three baselines: first, an i.i.d. sampling scheme that optimizes Problem (3) using the truncated loss (TRUNC-CVAR); second, an i.i.d. sampling scheme that uses a smoothing technique to relax the $\sum_i [x_i]_+$ non-linearity (SOFT-CVAR). Tarnopolskaya and Zhu (2010) compare different smoothing techniques for the $\sum_i [x_i]_+$ non-linearity. Of these, we use the relaxation $T \log(\sum_i e^{x_i/T})$ proposed by Nemirovski and Shapiro (2006). In each iteration, we heuristically approximate the population sum with a mini-batch. Third, we also compare a standard i.i.d. sampling ERM scheme that stochastically minimizes the average of the losses (MEAN).

### 5.1   Convex CVaR Optimization

We first compare the different algorithms in a controlled convex setting, where the classical TRUNC-CVAR algorithm is expected to perform well. We consider three UCI regression data sets, three synthetic regression data sets, and eight different UCI classification data sets (Dua and Graff, 2017). The left and middle plots in Figure 2 present a summary of the results (see Table 1 in Appendix E for a detailed version). We evaluate the CVaR ($\alpha = 0.01$) and average loss for linear regression and the accuracy, and CVaR and average surrogate loss for classification (logistic regression) on the test set. In linear regression, ADA-CVAR performs comparably or better to benchmarks in terms of the CVaR of the loss and is second best in terms of the average loss. In classification, TRUNC-CVAR performs better in terms of the CVaR for the *surrogate loss* but performs *poorly* in terms of accuracy. This is due to the fact that it learns a predictive distribution that is close to uniform. ADA-CVAR has a comparable accuracy to ERM (MEAN algorithm) but a much better CVaR. Hence, it finds a good predictive model while successfully controlling the prediction risk.

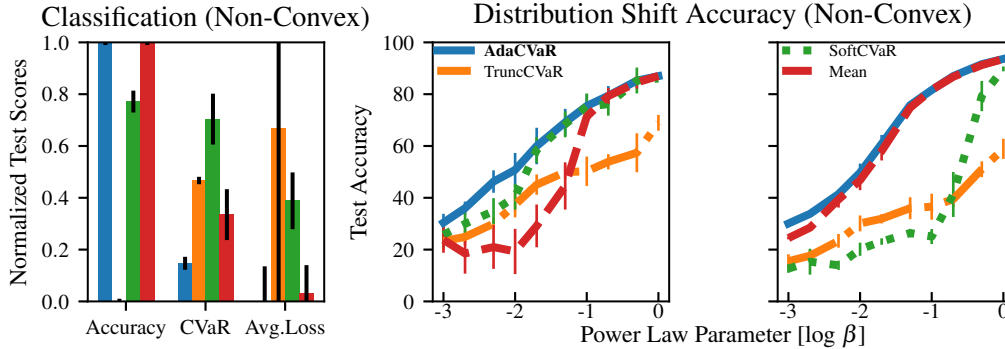

Figure 3: Non Convex Optimization tasks. *Left*: Normalized scores in image classification tasks. ADA-CVAR attains state-of-the-art accuracy and lowest CVaR. *Middle and Right*: Test accuracy under train/test distribution shift on CIFAR-10 for VGG16-BN (middle) and ResNet-18 (right). Lower $\beta$ indicates larger shift. ADA-CVAR has always better test accuracy than benchmarks.

## 5.2 Convex CVaR Distributional Robustness

We use the same classification data sets and classifiers as in Section 5.1. To produce the distribution shift, we randomly sub-sample the majority class in the training set, so that the new training set has a 10%/90% class imbalance and the majority/minority classes are inverted. The test set is kept unchanged. Such shifts in class frequencies are common (Mehrabi et al., 2019, Section 3.2).

We consider $\alpha = 0.1$, which is compatible with the data imbalance. The right plot in Figure 2 shows a summary of the results (See Table 2 in Appendix E for detailed results). ADA-CVAR has higher test accuracy than the benchmarks and is comparable to TRUNC-CVAR on average log-likelihood.

We note that the CVaR provides robustness with respect to worst-case distribution shifts. Such a *worst case* distribution might be too pessimistic to be encountered in practice, however. Instead, ADA-CVAR appears to benefit from the varying distributions during training and protects better against non-adversarial distribution shifts. Other techniques for dealing with imbalanced data might also be useful to address this distribution shift empirically, but are only useful if there is an a-priori knowledge of the class ratio in the test set. Instead, the CVaR optimization guards against any distribution shift. Furthermore, with such a-priori knowledge, such techniques can also be used together with ADA-CVAR. We provide extended experiments analyzing distribution shift in Appendix E.1.

## 5.3 Non-Convex (Deep Learning) CVaR Optimization

We test our algorithm on common non-convex optimization benchmarks in deep learning (MNIST, Fashion-MNIST, CIFAR-10). As it is common in these setting, we perform data-augmentation on the training set. Thus, the effective training set size is infinite. To address this, we consider a mixture of distributions in a similar spirit as Borsos et al. (2019). Each data point serves as a representative of a distribution over all its possible augmentations. We optimize the CVaR of this mixture of distributions as a surrogate of the CVaR of the infinite data set. The left plot in Figure 3 summarizes the results (See Table 3 in Appendix E). ADA-CVAR reaches the same accuracy as ERM in all cases and has lower CVaR. Only in CIFAR-10 it does not outperform TRUNC-CVAR in terms of the CVaR of the surrogate loss. This is because the TRUNC-CVAR yields a predictive model that is close to uniform. Instead, ADA-CVAR still yields useful predictions while controlling the CVaR.

**Gradient Magnitude and Training Time** The gradients of TRUNC-CVAR are either 0 or $1/\alpha$ times larger than the gradients of the same point using MEAN. A similar but smoothed phenomenon arises with SOFT-CVAR. This makes training these losses considerably harder due to exploding gradients and noisier gradient estimates. With the same learning rates, these algorithms usually produce numerical overflows and, to stabilize learning, we used considerably smaller learning rates. In turn, this increased the number of iterations required for convergence. ADA-CVAR does not suffer from this as the gradients have the same magnitude as in MEAN. For example, to reach 85 % *train* accuracy

ADA-CVAR requires 7 epochs, MEAN 9, SOFT-CVAR 21, and TRUNC-CVAR never surpassed 70 % train accuracy. There was no significant difference between time per epoch of each of the algorithms.

### 5.4 Distributional Robustness in Deep Learning through Optimizing the CVaR

Lastly, we demonstrate that optimizing the CVaR yields improved robustness to distribution shifts in deep learning. We simulate distribution shift through mismatching training and test class frequencies. Since we consider multi-class problems, we simulate power-law class frequencies, which are commonly encountered in various applications (Clauset et al., 2009). More specifically, we sub-sample each class of the training set of CIFAR-10 so that the class size follows a power-law distribution $p(|c|) \propto c^{\log \beta}$, where $|c|$ is the size of the $c$-th class and keep the test set unchanged. In middle and right plots of Figure 3, we show the test accuracy for different values of $\beta$ for VGG16-BN and ResNet-18 networks. The algorithms do not know a-priori the amount of distribution shift to protect against and consider a fixed $\alpha = 0.1$. For all distribution shifts, ADA-CVAR is superior to the benchmarks.

When a high-capacity network learns a perfectly accurate model, then the average and CVaR of the loss distribution have both zero value. This might explain the similarity between MEAN and ADA-CVAR for ResNet-18. Instead, there is a stark discrepancy between ADA-CVAR and MEAN in VGG16. This shows the advantage of training in a risk averse manner, particularly when the model makes incorrect predictions due to a strong inductive bias.

## 6 Conclusions

The CVaR is a natural criterion for training ML models in a risk-aware fashion. As we saw, the traditional way of optimizing it via truncated losses fails for modern machine learning tasks due to high variance of the gradient estimates. Our novel adaptive sampling algorithm ADA-CVAR exploits the distributionally robust optimization formulation of the CVaR, and tackles it via regret minimization. It naturally enables the use of standard stochastic optimization approaches (e.g., SGD), applied to the marginal distribution of a certain k-DPP. Finally, we demonstrate in a range of experiments that ADA-CVAR is superior to the TRUNC-CVAR algorithm for regression and classification tasks, both in convex and non-convex learning settings. Furthermore, ADA-CVAR provides higher robustness to (non-adversarial) distribution shifts than TRUNC-CVAR, SOFT-CVAR, or MEAN algorithms.

## Broader Impact

Increasing *reliability* is one of the central challenges when deploying machine learning in high-stakes applications. We believe our paper makes important contributions to this endeavor by going beyond simply optimizing the *average* performance, and considering risk in deep learning. The CVaR is also known to be an avenue towards enforcing *fairness constraints* in data sets (Williamson and Menon, 2019). Hence, our algorithm also contributes to optimizing fair deep models, by counteracting inherent biases in the data (e.g., undersampling of certain parts of the population).

## Acknowledgments and Disclosure of Funding

This project has received funding from the European Research Council (ERC) under the European Unions Horizon 2020 research, innovation programme grant agreement No 815943, a DARPA YFA award, and NSF CAREER award 1553284.

## Footnotes

[1]Note that we do *not* use any importance sampling correction.

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
