[Supplementary Material]

# A   Approximate Sampling from k-DPP Marginals

The final piece of the ADA-CVAR algorithm is how to efficiently compute the marginal distribution $\mathbb{P}_w(i)$ of the k-DPP model, and how to sample from it.

**Cost of sampling**   Compared to general k-DPPs, our setting has the crucial advantage that the kernel matrix is diagonal. Thus there is no need for performing an expensive eigendecomposition of the kernel matrix which is required in the general case. The marginals of diagonal k-DPPs are

$$\mathbb{P}_w(i) = w_i \frac{e_{-i}^{k-1}}{e_N^k}, \tag{9}$$

where $e_N^k = \sum_{|I|=k} \prod_{i \in I} w_i$ is the elementary symmetric polynomial of size $k$ for the ground set $[N]$ and $e_{-i}^{k-1}$ is the elementary symmetric polynomial of size $k-1$ for the ground set $[N] \setminus i$. Unfortunately, naively computing the elementary symmetric polynomials has a complexity of $O(N^2 k) = O(\alpha N^3)$ using (Kulesza et al., 2012, Algorithm 7). Even if this computation could be performed fast, exact computation of the elementary symmetric polynomials is numerically unstable.

In view of this, Barthelmé et al. (2019) propose an approximation to k-DPPs valid for large-scale ground sets which has better numerical properties. Their main idea is to relax the sample size constraint of the k-DPP with a soft constraint such that the *expected* sample size of the matched DPP is $k$. The total variation distance between the marginal probabilities of the k-DPP and DPP decays as $O(1/N)$. The marginal probabilities of this matched DPP are simply

$$\hat{\mathbb{P}}_w(i) = \frac{w_i e^\nu}{1 + w_i e^\nu}, \tag{10}$$

where $\nu$ softly enforces the sample size constraint $\sum_{i=1}^N \frac{w_i e^\nu}{1 + w_i e^\nu} = k$. For a given $\nu$, we can efficiently sample from this singleton-marginal distribution, which takes $O(\log(N))$ using the same sum-tree data-structure as Shalev-Shwartz and Wexler (2016).

**Proposition 2** (Excess Regret of Approximate Sampler). *Let $\tilde{q}$ be the marginals of the approximate k-DPP* (10) *and $q$ the exact marginals of the k-DPP* (9). *Then, a sampler that plays* ADA-CVAR *using the approximate k-DPPs marginals suffers an extra regret of $\epsilon_{\text{approx}} T$, with $\epsilon_{\text{approx}} = O(1/N)$, for large enough $N$.*

*Proof.* Let $\widetilde{\text{SR}}_T$ be the regret of the player that selects the approximate k-DPPs marginals. Then,

$$\widetilde{\text{SR}}_T := \max_{q \in \mathcal{Q}^\alpha} \sum_{t=1}^T q^\top L_t - \sum_{t=1}^T \tilde{q}_t^\top L_t$$

$$= \text{SR}_T + \sum_{t=1}^T (\tilde{q}_t - q_t)^\top L_t$$

$$\leq \text{SR}_T + \sum_{t=1}^T \|\tilde{q}_t - q_t\|_1 \|L_t\|_\infty$$

$$\leq \text{SR}_T + \sum_{t=1}^T \|\tilde{q}_t - q_t\|_1$$

$$\leq \text{SR}_T + \epsilon_{\text{approx}} T$$

where the first inequality holds due to Hölder's inequality, the second inequality due to $L \in [0,1]$, and finally noticing that $\|\tilde{q}_t - q_t\|_1$ is proportional to the total variation distance. Using Theorem 2.1 from Barthelmé et al. (2019) the results follow. $\square$

We note that this result implies an an extra $O(1/N)$ term in the excess CVaR bounds from Corollaries 1 and 2. This is fast compared to statistical $O(1/\sqrt{N})$ rates. For small $N$, the exact marginals can be computed efficiently so there is no need for the approximation.

# B Omitted Proofs

## B.1 Proof of Proposition 1

**Proposition 1.** Let $h : \mathcal{X} \to \mathcal{Y}$ be a function class with finite VC-dimension $|\mathcal{H}|$. Let $L(h) : \mathcal{H} \to [0,1]$ be a random variable. Then, for any $0 < \alpha \leq 1$, with probability at least $1 - \delta$,

$$\mathbb{E}\left[\sup_{h \in \mathcal{H}} \left|\widehat{\mathbb{C}}^{\alpha}[L(h)] - \mathbb{C}^{\alpha}[L(h)]\right|\right] \leq \frac{1}{\alpha}\sqrt{\frac{\log(2|\mathcal{H}|/\delta)}{N}}.$$

*Proof for Proposition 1.* Brown (2007) proves that the following two inequalities hold jointly with probability $1 - \delta$, $\delta \in (0,1]$ for a single $h \in \mathcal{H}$:

$$\mathbb{C}^{\alpha}(L(h)) \geq \widehat{\mathbb{C}}^{\alpha}(L(h)) - \frac{1}{\alpha}\sqrt{\frac{\log(2/\delta)}{N}}$$

$$\mathbb{C}^{\alpha}(L(h)) \leq \widehat{\mathbb{C}}^{\alpha}(L(h)) + \sqrt{\frac{5\log(6/\delta)}{\alpha N}}$$

Taking the union bound over all $h \in \mathcal{H}$:

$$\mathbb{C}^{\alpha}(L(h)) \geq \widehat{\mathbb{C}}^{\alpha}(L(h)) - \frac{1}{\alpha}\sqrt{\frac{\log(2|\mathcal{H}|/\delta)}{N}}$$

$$\mathbb{C}^{\alpha}(L(h)) \leq \widehat{\mathbb{C}}^{\alpha}(L(h)) + \sqrt{\frac{5\log(6|\mathcal{H}|/\delta)}{\alpha N}}$$

The theorem follows from taking the maximum between lower and upper bounds.

If $\mathcal{H}$ is a non-finite class we follow a standard argument with bounded norm of real valued functions, i.e., $\|\mathcal{H}\|_{\infty} \leq B$, we rely on the standard argument based on covering numbers. To define such covering, we fix $\epsilon > 0$ and consider a set $\mathcal{C}_{\mathcal{H},\epsilon}$ of minimum cardinality, such that for all $h \in \mathcal{H}$, there exists an $h' \in \mathcal{C}_{\mathcal{H},\epsilon}$, satisfying $|h - h'| \leq \epsilon$. Define the covering number as $\mathcal{N}_{\mathcal{H},\epsilon} = |\mathcal{C}_{\mathcal{H},\epsilon}|$, then taking the union bound for $\epsilon = 1\sqrt{N}$ we arrive to the result

$$\mathbb{E}\left[\sup_{h \in \mathcal{H}} \left|\widehat{\mathbb{C}}^{\alpha}[L(h)] - \mathbb{C}^{\alpha}[L(h)]\right|\right] \leq \frac{1}{\alpha}\sqrt{\frac{\log(2\mathcal{N}_{\mathcal{H},\epsilon}/\delta)}{N}}.$$

Finally, the logarithm of the covering number can be upper bounded using the pseudo-VC dimension of the function class, for a proof see Devroye et al. (2013, Chapter 29). □

## B.2 Proof of Lemma 1

**Lemma 1.** Let the sampler player play the к.EXP.3 Algorithm with $\eta = \sqrt{\frac{\log N}{NT}}$. Then, she suffers a sampler regret (5) of at most $O(\sqrt{TN\log N})$.

In order to prove this, we need to first show that к.EXP.3 is a valid algorithm for the sampler player. This we do next.

**Proposition 3.** *The marginals of any k-DPP with a diagonal matrix kernel $K = \mathrm{diag}(w)$ are in the set $\mathcal{Q}_k^{\alpha} = \left\{ kq \in \mathbb{R}^N \mid q \in \mathcal{Q}^{\alpha} \right\}$.*

*Proof.* For any $w \in \mathbb{R}_{\geq 0}^N$ the marginals of the k-DPP with kernel $K = \mathrm{diag}(w)$ are:

$$\mathbb{P}_w(i) = \sum_{I \ni i} \mathbb{P}_w(I) = \frac{\sum_{I \ni i} \prod_{i' \in I} w_{i'}}{\sum_I \prod_{i' \in I} w_{i'}}. \tag{11}$$

From eq. (11), clearly $0 \leq \mathbb{P}_w(i) \leq 1$. Summing eq. (11) over $i$ we get:

$$\sum_i \mathbb{P}_w(i) = \sum_i \sum_{I \ni i} \mathbb{P}_w(I) = \sum_I \mathbb{P}_w(I) \sum_{i \in I} 1 = k$$

This shows that $\mathbb{P}_w(i) \in \mathcal{Q}_k^{\alpha}$. □

**Proposition 4.** *Let* $\tilde{L}_I = \sum_{i \in I} L_i$. *Let* $\Delta = \left\{ \tilde{w} \in \mathbb{R}^{\binom{N}{k}} \mid, 0 \le w_I \le 1, \sum_I w_I = 1 \right\}$ *the set of distributions over the* $\binom{N}{k}$ *subsets of size $k$ of the ground set $[N]$.*

$$\max_{q \in \mathcal{Q}^\alpha} \sum_{t=1}^T q^\top L_t = \max_{\tilde{w} \in \Delta} \frac{1}{k} \sum_{t=1}^T \tilde{w} \tilde{L}_I \tag{12}$$

*Proof.* Both left and right sides of (12) are linear programs over a convex polytope, hence the solution is in one of its vertices (Murty, 1983). The vertices of $\mathcal{Q}^\alpha$ are vectors $\frac{1}{k}\mathbf{1}_I = \frac{1}{k}[[i \in I]]$. These vectors have $\frac{1}{k}$ in coordinate $i$ if the coordinate belongs to set $I$ and 0 otherwise. The vertices of the simplex are just $[[I]]$, one for coordinate $I$.

Let $q^\star = \frac{1}{k}\mathbf{1}_{I^\star}$ be the solution of the l.h.s. of (12). Assume that $\hat{I} \ne I^\star$ is the solution of the right hand side. This implies that $\tilde{L}_{\hat{I}} \ge \tilde{L}_{I^\star}$. Therefore, $\sum_{i \in \hat{I}} L_i \ge \sum_{i \in I^\star} L_i$. This in turn implies that $\mathbf{1}_{\hat{I}} L_i \ge \mathbf{1}_{I^\star} L_i$, which contradicts the first predicate. In the case the equalities hold, then the values l.h.s and r.h.s. of equation (12) are also equal. $\square$

*Proof of Lemma 1.*

$$\begin{aligned}
\mathrm{SR}_T &= \max_{q \in \mathcal{Q}^\alpha} \sum_{t=1}^T q^\top L_t - \sum_{t=1}^T q_t^\top L_t \\
&= \max_{\tilde{w} \in \Delta} \frac{1}{k} \sum_{t=1}^T \tilde{w} \tilde{L}_I - \sum_{t=1}^T q_t^\top L_t \\
&= \frac{1}{k} \left( \max_{\tilde{w} \in \Delta} \sum_{t=1}^T \tilde{w} \tilde{L}_I - \sum_{t=1}^T \sum_i \mathbb{P}_{w_t}(i) L_i \right) \\
&= \frac{1}{k} \left( \max_{\tilde{w} \in \Delta} \sum_{t=1}^T \tilde{w} \tilde{L}_I - \sum_{t=1}^T \sum_I \mathbb{P}_{w_t}(I) \tilde{L}_I \right) \\
&\le O(\sqrt{NT \log(N)}) \qquad\qquad \square
\end{aligned}$$

The first equality uses Proposition 4. The second equality uses Proposition 3 and the fact that the iterates $q_t$ come from the K.EXP.3 algorithm. The third equality uses the definition of $\tilde{L}$. The final inequality is due to Alatur et al. (2020, Lemma 1).

### B.3 Proof of Lemma 2

**Lemma 2.** *A learner player that plays the BTL algorithm with access to an ERM oracle as in Assumption 1, achieves at most $\epsilon_{\mathrm{oracle}} T$ regret.*

*Proof.* We proceed by induction. Clearly for $T = 1$, $\mathrm{LR}_1 \le \epsilon_{\mathrm{oracle}}$. Assume true for $T-1$, the inductive hypothesis is $\mathrm{LR}_{T-1} \le \epsilon_{\mathrm{oracle}}(T-1)$. The regret at time $T$ is:

$$\begin{aligned}
\mathrm{LR}_T &= \sum_{t=1}^T q_t^\top L(\theta_t) - \min_{\theta \in \Theta} \sum_{t=1}^T q_t^\top L(\theta), \\
&\le \sum_{t=1}^T q_t^\top \left[ L(\theta_t) - L(\theta_T) \right] + \epsilon_{\mathrm{oracle}} \\
&= \sum_{t=1}^{T-1} q_t^\top \left[ L(\theta_t) - L(\theta_T) \right] + \epsilon_{\mathrm{oracle}} \\
&\le \mathrm{LR}_{T-1} + \epsilon_{\mathrm{oracle}} \le \epsilon_{\mathrm{oracle}} T \qquad\qquad \square
\end{aligned}$$

The first inequality is due to the definition of the oracle, the second inequality is by definition of the minimum in the learner regret, and the final inequality is due to the inductive hypothesis.

## B.4   Proof of Theorem 1

**Theorem 1.** Let $L_i(\cdot) : \Theta \to [0,1]$, $i = \{1, ..., N\}$ be a fixed set of loss functions. If the sampler player uses ADA-CVAR and the learner player uses the BTL algorithm, then the game has regret $O(\sqrt{TN \log N})$.

*Proof.* We decompose bound the game regret into the sum of player and sampler regret. To bound the regret, we bound it with the sum of the Learner and Sampler regret as follows:

$$\text{GameRegret}_T = \sum_{t=1}^{T} J(\theta_t, q^\star) - J(\theta^\star, q_t),$$

$$\leq \max_{q \in \mathcal{Q}^\alpha} \sum_{t=1}^{T} J(\theta_t, q) - J(\theta^\star, q_t),$$

$$(\text{Lemma 2}) \leq \max_{q \in \mathcal{Q}^\alpha} \sum_{t=1}^{T} J(\theta_t, q) - J(\theta_t, q_t),$$

$$(\text{Lemma 1}) \leq O(\sqrt{TN \log N} + \epsilon_{\text{oracle}} T). \qquad \square$$

## B.5   Proof of Corollary 1

**Corollary 1** (Online to Batch Conversion)**.** Let $L_i(\cdot) : \Theta \to [0,1]$, $i = \{1, ..., N\}$ be a set of loss functions sampled from a distribution $\mathcal{D}$. Let $\theta^\star$ be the minimizer of the CVaR of the empirical distribution $\hat{\mathbb{C}}^\alpha$. Let $\bar{\theta}$ be the output of ADA-CVAR, selected uniformly at random from the sequence $\{\theta_t\}_{t=1}^T$. Its expected excess CVaR is bounded as:

$$\mathbb{E}\hat{\mathbb{C}}^\alpha[L(\bar{\theta})] \leq \hat{\mathbb{C}}^\alpha[L(\theta^\star)] + +O(\sqrt{N \log N / T}) + \epsilon_{\text{oracle}}$$

where the expectation is taken w.r.t. the randomization in the algorithm, both for the sampling steps and the final randomization in choosing $\bar{\theta}$.

*Proof.* We use the proof techinque from Agarwal et al. (2018)[Theorem 3]. The CVaR of $\bar{\theta}$ is $\hat{\mathbb{C}}^\alpha[L(\bar{\theta})] = \max_{q \in \mathcal{Q}^\alpha} J(\bar{\theta}, q)$. Let $\bar{q}^\star := \arg\max_{q \in \mathcal{Q}^\alpha} J(\bar{\theta}, q)$. For any $q \in \mathcal{Q}^\alpha$, we can use the sampler regret to bound:

$$\mathbb{E}\left[J(\bar{\theta}, q)\right] = \mathbb{E}\left[\frac{1}{T}\sum_{t=1}^{T} J(\theta_t, q)\right]$$

$$\leq \mathbb{E}\left[\frac{1}{T}\sum_{t=1}^{T} J(\theta_t, q_t)\right] + \frac{1}{T}\text{SR}_T$$

Likewise, for any $\theta \in \Theta$,

$$\mathbb{E}\left[J(\theta, \bar{q})\right] = \mathbb{E}\left[\frac{1}{T}\sum_{t=1}^{T} J(\theta, q_t)\right]$$

$$\geq \mathbb{E}\left[\frac{1}{T}\sum_{t=1}^{T} J(\theta_t, q_t)\right] - \frac{1}{T}\text{LR}_T$$

Using the minimax inequality, we bound the excess risk for any $q \in \mathcal{Q}$ and $\theta \in \Theta$ by the average game regret:

$$\mathbb{E}\left[J(\bar{\theta}, q)\right] \leq J(\theta^\star, q^\star) + \mathbb{E}\left[J(\bar{\theta}, q) - J(\theta, \bar{q})\right]$$

$$\leq J(\theta^\star, q^\star) + \underbrace{\frac{1}{T}\left(\text{LR}_T + \text{SR}_T\right)}_{\epsilon}$$

Noting that, $\text{GameRegret}_T = \text{LR}_T + \text{SR}_T$ and that the latter results holds in particular for $\bar{q}^\star$. Furthermore, the pair $(\bar{\theta}, \bar{q})$ is an $\epsilon$-equilibrium point of the game in the sense: $J(\theta^\star, q^\star) - \epsilon \leq \mathbb{E}\left[J(\bar{\theta}, \bar{q})\right] \leq J(\theta^\star, q^\star) + \epsilon$. $\qquad \square$

## B.6 Proof of Corollary 2

**Corollary 2** (Online to Batch Conversion). Let $L(\cdot) : \Theta \to [0, 1]$ be a Random Variable induced by the data distribution $\mathcal{D}$. Let $\theta^\star$ be the minimizer of the CVaR at level $\alpha$ of such Random Variable. Let $\bar{\theta}$ be the output of ADA-CVAR, selected uniformly at random from the sequence $\{\theta_t\}_{t=1}^T$. Then, with probability at least $\delta$ the expected excess CVaR of $\bar{\theta}$ is bounded as:

$$\mathbb{E}\mathbb{C}^\alpha[L(\bar{\theta})] \leq \mathbb{C}^\alpha[L(\theta^\star)] + O(\sqrt{N\log N/T}) + \epsilon_{\text{oracle}} + \epsilon_{\text{stat}}$$

where $\epsilon_{\text{stat}} = \tilde{O}(\frac{1}{\alpha}\sqrt{\frac{1}{N}})$ comes from the statistical error and the expectation is taken w.r.t. the randomization in the algorithm, both for the sampling steps and the randomization in choosing $\bar{\theta}$.

*Proof.*

$$\begin{aligned}
\mathbb{E}\mathbb{C}^\alpha[L(\bar{\theta})] &\leq \mathbb{E}\hat{\mathbb{C}}^\alpha[L(\bar{\theta})] + \epsilon_{\text{stat}} \\
&\leq \mathbb{E}\hat{\mathbb{C}}^\alpha[L(\theta^\star)] + O(\sqrt{N\log N/T}) + \epsilon_{\text{oracle}} + \epsilon_{\text{stat}} \\
&\leq \mathbb{E}\mathbb{C}^\alpha[L(\theta^\star)] + O(\sqrt{N\log N/T}) + \epsilon_{\text{oracle}} + 2\epsilon_{\text{stat}},
\end{aligned}$$

Where the first and last inequality hold by the uniform convergence results from Proposition 1 and the second inequality by Corollary 1. $\qquad\square$

## C  Learner Player Algorithm for Convex Losses

In the convex setting, there are online learning algorithms that have no-regret guarantees and there is no need to play the BTL algorithm (7) exactly. Instead, stochastic gradient descent (SGD) Zinkevich (2003) or online mirror descent (OMD) (Beck and Teboulle, 2003) both have no-regret guarantees. We focus now on SGD but, for certain geometries of $\Theta$ and appropriate mirror maps, OMD has exponentially better regret guarantees (in terms of the dimension of the problem).

---

**Algorithm 2:** Ada-CVaR-CVX

---

**input** Learning rates $\eta_s, \eta_l$.
 1: **Sampler:** Initialize k-DPP $w_1 = \mathbf{1}_N$.
 2: **Learner:** Initialize parameters $\theta_1 \in \Theta$.
 3: **for** $t = 1, \ldots, T$ **do**
 4:    **Sampler:** Sample element $i_t \sim q_t = \frac{1}{k}\mathbb{P}_{w_t}(i)$.
 5:    **Sampler**: Build estimate $\hat{L}_t = \frac{L_{i_t}(\theta_t)}{q_{t,i_t}}[[i == i_t]]$.
 6:    **Sampler**: Update k-DPP $w_{t+1,i} = w_{t,i}e^{\eta_s \hat{L}_{t,i}}$.
 7:    **Learner:** $\theta_{t+1} = \theta_t - \eta_l \nabla L_{i_t}(\theta_t)$.
 8: **end for**
**output** $\bar{\theta} = \frac{1}{T}\sum_{t=1}^T \theta_t$, $\bar{q} = \frac{1}{T}\sum_{t=1}^T q_t$

---

**Lemma 3.** *Let assume that $L_i(\cdot) : \Theta \to [0, 1]$ be any sequence of convex losses, with $\|\nabla L_i\|_2 \leq G$ and $\|\Theta\|_2 \leq D$, then a learner player that plays SGD algorithm suffers at most regret $O(GD\sqrt{T})$.*

*Proof.* Hazan (2016, Chapter 3). $\qquad\square$

Note that even if there are algorithms for the strongly convex case or exp-concave case that have $\log(T)$ regret, it does not bring any advantage in our case as the $\sqrt{T}$ term in the sampler regret dominates and is unavoidable (Audibert et al., 2013).

**Corollary 3.** *Let $L_i(\cdot) : \Theta \to [0, 1]$ be any sequence of convex losses. Let the learner and sampler player play Algorithm 2, then the game has regret $O(\sqrt{TN\log N} + \beta\sqrt{T})$, where $\beta$ is a problem-dependent constant.*

This result changes the $\epsilon_{\text{oracle}}$ term in Corollaries 1 and 2 by a $O(1/\sqrt{T})$ term.

# D  Experimental Setup

**Implementation and Resources:**  We implemented all our experiments using PyTorch (Paszke et al., 2017). We ran our experiments convex experiments on an Intel(R) Xeon(R) CPU E5-2697 v4 2.30GHz machine. Our deep lerning experiments ran on an NVIDIA GeForce GTX 1080 Ti GPU.

**Datasets:**  For classification we use the Adult, Australian Credit Approval, German Credit Data, Monks-Problems-1, Spambase, and Splice-junction Gene Sequences datasets from the UCI repository (Dua and Graff, 2017) and the Titanic Disaster dataset from (Eaton and Haas, 1995). For regression we use the Boston Housing, and Abalone, and CPU small from the UCI repository, the sinc dataset is synthetic recreated from (Fan et al., 2017), and normal and pareto datasets are synthetic datasets recreated from (Brownlees et al., 2015) with Gaussian and Pareto noise, respectively. For vision datasets, we use MNIST (LeCun et al., 1998), Fashion-MNIST (Xiao et al., 2017), and CIFAR-10 (Krizhevsky et al., 2014).

**Models:**  For convex regression and classification models we use linear models. For MNIST we use LeNet-5 neural network (LeCun et al., 1995). For Fashion MNIST we use LeNet-5 using dropout (Hinton et al., 2012). For CIFAR-10 we use ResNet18 (He et al., 2016) neural network and VGG-16 (Simonyan and Zisserman, 2014) with batch normalization (Ioffe and Szegedy, 2015).

**Dataset Preparation:**  We split UCI datasets into train, validation, and test set using a 50/30/20 split. For vision tasks we use as validation set the same images in the train set, without applying data-augmentations. For discrete categorical data, we use a one-hot-encoding. We standarize continuous data.

**Hyper-Parameter Search:**  We ran a grid search over the hyperparameters for all the algorithms with a five different random seeds. In regression tasks, we selected the set of hyper-parameters with the lowest CVaR in the validation set. In classification tasks, we selected the set of hyper-parameters with the highest accuracy in the validation set. The hyperparameters are:

1. Optimizer SGD with momentum or ADAM (Kingma and Ba, 2014).
2. Initial learning rates: $\{0.05, 0.01, 0.005, 0.001, 0.0005, 0.0001, 0.00001\}$,
3. Momentum: 0.9
4. Learning rate decay: 0.1 at epochs 20 and 40.
5. Batch Size $\{64, 128\}$,
6. Adaptive algorithm learning rate: $\{1.0, 0.5, 0.1, *\}$, where (*) is the optimal learning rate,
7. Mixing with uniform distribution: $\{0, 0.01, 0.1\}$,
8. Adaptive algorithm learning rate decay scheduling: $\{\text{constant}, O(1/\sqrt{t}), \text{Adagrad}\}$,
9. SOFT-CVAR algorithm temperature: $\{0.1, 1, 10\}$.
10. Random seeds: $\{0, 1, 2, 3, 4\}$.
11. Early stopping: $\{\text{True}, \text{False}\}$.

**Experimental Significance:**  In UCI and synthetic datasets, the test results of each algorithm is paired because the same data split is used across the algorithms. Likewise, for vision datasets, the neural network initialization is paired across experiments. Therefore, we use a paired t-test to determine statistical significance for $p \leq 0.05$ (Zimmerman, 1997). In TRUNC-CVAR and SOFT-CVAR experiments, we usually encountered numerical overflows, we discarded such experiments to determine significance.

**Evaluation Metric Normalization**  We evaluate the CVaR and the Average Loss for regression and we add Accuracy, and the CVaR and the Average Loss of the surrogate loss for classification. We normalize the mean score $s$ of algorithm $a$ on a given task by $(s_a - \min_a s_a)/(\max_a s_a - \min_a s_a)$ and we divide the standard deviation by $\max_a s_a$.

# E  Extended Experimental Results

In Table 1 we show the results of Section 5.1, in Table 2 the results of Section 5.2, and in Table 3 the results of Section 5.3.

Table 1: Test mean $\pm$ s.d. over five independent data splits. In shaded bold we indicate the best algorithms. In regression, we show the CVaR/(mean) loss. ADA-CVAR is competitive to benchmarks optimizing the CVaR. In classification, we show the accuracy / precision in "Accuracy" rows and CVaR/loss in "Surrogate" rows. ADA-CVAR has similar accuracy to MEAN/SOFT-CVAR, but with a lower CVaR.

| | Data Set | ADA-CVAR | | TRUNC-CVAR | | SOFT-CVAR | | MEAN | |
|---|---|---|---|---|---|---|---|---|---|
| **Regression Loss** | Abalone | **8.92 ± 3.3** | 0.61 ± 0.1 | **7.94 ± 2.7** | 0.79 ± 0.1 | 14.25 ± 0.3 | 0.63 ± 0.0 | 11.07 ± 3.7 | **0.51 ± 0.1** |
| | Boston | **3.09 ± 0.8** | 0.28 ± 0.1 | **3.02 ± 1.0** | 0.36 ± 0.0 | 3.28 ± 1.4 | **0.24 ± 0.1** | 4.51 ± 1.9 | **0.27 ± 0.1** |
| | Cpu | 2.32 ± 0.2 | 0.58 ± 0.0 | **2.12 ± 0.2** | 0.57 ± 0.1 | 2.85 ± 0.0 | 0.40 ± 0.0 | 9.95 ± 1.4 | **0.30 ± 0.0** |
| | Normal | 0.22 ± 0.1 | 0.03 ± 0.0 | 0.42 ± 0.3 | 0.06 ± 0.0 | **0.18 ± 0.0** | **0.01 ± 0.0** | 0.55 ± 0.2 | 0.07 ± 0.0 |
| | Pareto | 0.39 ± 0.3 | 0.02 ± 0.0 | 0.43 ± 0.1 | 0.05 ± 0.0 | **0.30 ± 0.3** | **0.01 ± 0.0** | 0.69 ± 0.1 | 0.06 ± 0.0 |
| | Sinc | 7.70 ± 3.1 | **0.80 ± 0.2** | 7.82 ± 3.5 | **0.74 ± 0.2** | 7.82 ± 3.5 | **0.72 ± 0.2** | 8.40 ± 3.9 | **0.71 ± 0.2** |
| **Classification Accuracy** | Adult | **0.85 ± 0.0** | 0.72 ± 0.0 | 0.71 ± 0.1 | 0.44 ± 0.2 | **0.85 ± 0.0** | **0.74 ± 0.0** | **0.85 ± 0.0** | **0.74 ± 0.0** |
| | Australian | **0.82 ± 0.0** | **0.79 ± 0.0** | 0.66 ± 0.1 | 0.62 ± 0.1 | **0.82 ± 0.0** | **0.81 ± 0.0** | 0.81 ± 0.0 | 0.78 ± 0.0 |
| | German | 0.74 ± 0.0 | 0.65 ± 0.0 | 0.64 ± 0.0 | 0.43 ± 0.0 | **0.78 ± 0.0** | **0.71 ± 0.0** | **0.77 ± 0.0** | **0.67 ± 0.1** |
| | Monks | **0.62 ± 0.1** | **0.59 ± 0.1** | 0.53 ± 0.1 | 0.50 ± 0.1 | **0.63 ± 0.0** | 0.68 ± 0.0 | 0.61 ± 0.1 | 0.66 ± 0.1 |
| | Phoneme | **0.75 ± 0.0** | **0.59 ± 0.0** | 0.47 ± 0.1 | 0.26 ± 0.0 | **0.75 ± 0.0** | **0.60 ± 0.0** | **0.76 ± 0.0** | **0.60 ± 0.0** |
| | Spambase | 0.90 ± 0.0 | 0.88 ± 0.0 | 0.81 ± 0.0 | 0.71 ± 0.0 | **0.93 ± 0.0** | **0.92 ± 0.0** | **0.92 ± 0.0** | 0.91 ± 0.0 |
| | Splice | **0.94 ± 0.0** | **0.93 ± 0.0** | 0.90 ± 0.0 | 0.89 ± 0.0 | 0.92 ± 0.0 | 0.91 ± 0.0 | 0.93 ± 0.0 | 0.92 ± 0.0 |
| | Titanic | **0.78 ± 0.0** | **0.74 ± 0.0** | 0.55 ± 0.2 | 0.40 ± 0.3 | **0.78 ± 0.0** | **0.75 ± 0.0** | **0.78 ± 0.0** | **0.75 ± 0.0** |
| **Classification Surrogate** | Adult | 1.95 ± 0.1 | 0.37 ± 0.0 | **0.70 ± 0.0** | 0.69 ± 0.0 | 3.08 ± 0.1 | **0.32 ± 0.0** | 3.13 ± 0.1 | 0.32 ± 0.0 |
| | Australian | 1.33 ± 0.1 | 0.49 ± 0.0 | **0.83 ± 0.0** | 0.66 ± 0.0 | 1.78 ± 0.1 | 0.47 ± 0.0 | 1.93 ± 0.1 | **0.45 ± 0.0** |
| | German | 1.41 ± 0.2 | 0.56 ± 0.0 | **0.86 ± 0.0** | 0.67 ± 0.0 | 2.04 ± 0.2 | **0.50 ± 0.0** | 1.90 ± 0.2 | **0.51 ± 0.0** |
| | Monks | **0.89 ± 0.0** | **0.66 ± 0.0** | **0.88 ± 0.0** | 0.68 ± 0.0 | 1.36 ± 0.1 | 0.69 ± 0.0 | 1.07 ± 0.0 | **0.66 ± 0.0** |
| | Phoneme | 0.83 ± 0.0 | 0.64 ± 0.0 | **0.69 ± 0.0** | 0.69 ± 0.0 | 2.56 ± 0.0 | **0.47 ± 0.0** | 2.63 ± 0.0 | **0.47 ± 0.0** |
| | Spambase | **1.04 ± 0.1** | 0.49 ± 0.0 | 1.15 ± 0.3 | 0.49 ± 0.0 | 6.30 ± 1.7 | **0.23 ± 0.0** | 5.23 ± 1.4 | **0.23 ± 0.0** |
| | Splice | 1.57 ± 0.2 | 0.27 ± 0.0 | **1.06 ± 0.1** | 0.49 ± 0.0 | 5.45 ± 0.9 | **0.20 ± 0.0** | 1.75 ± 0.2 | **0.22 ± 0.0** |
| | Titanic | 0.78 ± 0.0 | 0.66 ± 0.0 | **0.71 ± 0.0** | 0.70 ± 0.0 | 1.70 ± 0.0 | **0.52 ± 0.0** | 1.68 ± 0.0 | **0.52 ± 0.0** |

Table 2: Test accuracy/loss (mean $\pm$ s.d.) over five independent data splits with *train/test distribution shift*. In shaded bold we indicate the best algorithms. ADA-CVAR has superior test accuracy than benchmarks. It has comparable test loss to TRUNC-CVAR.

| | Data Set | ADA-CVAR | | TRUNC-CVAR | | SOFT-CVAR | | MEAN | |
|---|---|---|---|---|---|---|---|---|---|
| **Distribution Shift 10%** | Adult | **0.65 ± 0.0** | **0.63 ± 0.0** | 0.61 ± 0.1 | 0.69 ± 0.0 | 0.63 ± 0.1 | 0.84 ± 0.3 | 0.60 ± 0.1 | 0.90 ± 0.4 |
| | Australian | **0.68 ± 0.0** | **0.58 ± 0.0** | 0.46 ± 0.1 | 0.70 ± 0.0 | 0.48 ± 0.3 | 0.91 ± 0.4 | 0.46 ± 0.3 | 0.87 ± 0.3 |
| | German | **0.48 ± 0.1** | **0.73 ± 0.0** | 0.44 ± 0.1 | 0.75 ± 0.0 | 0.45 ± 0.1 | 1.00 ± 0.2 | **0.48 ± 0.1** | 0.80 ± 0.1 |
| | Monks | **0.49 ± 0.0** | 0.72 ± 0.0 | 0.39 ± 0.1 | 0.73 ± 0.0 | 0.43 ± 0.3 | 1.27 ± 0.6 | 0.40 ± 0.2 | 0.91 ± 0.3 |
| | Phoneme | **0.58 ± 0.1** | **0.68 ± 0.0** | 0.52 ± 0.2 | 0.69 ± 0.0 | 0.51 ± 0.2 | 1.08 ± 0.4 | 0.55 ± 0.2 | 0.92 ± 0.3 |
| | Spambase | **0.77 ± 0.0** | **0.46 ± 0.0** | 0.74 ± 0.1 | 0.60 ± 0.0 | 0.71 ± 0.2 | 0.61 ± 0.2 | 0.69 ± 0.2 | 0.58 ± 0.2 |
| | Splice | **0.84 ± 0.0** | **0.40 ± 0.0** | 0.77 ± 0.1 | 0.57 ± 0.0 | 0.73 ± 0.1 | 0.52 ± 0.2 | 0.50 ± 0.3 | 0.81 ± 0.4 |
| | Titanic | 0.46 ± 0.2 | **0.70 ± 0.0** | **0.50 ± 0.3** | **0.69 ± 0.0** | 0.43 ± 0.3 | 1.17 ± 0.5 | 0.41 ± 0.3 | 0.87 ± 0.3 |

Table 3: "Accuracy" rows: Avg. test accuracy/(min class) precision. "Surrogate" rows: Avg. test CVaR/loss. Average over 5 different random seeds. ADA-CVAR matches the accuracy and average loss performance of MEAN, but has a lower CVaR. TRUNC-CVAR and SOFT-CVAR struggle to learn in non-convex tasks.

| | Data Set | ADA-CVAR | | TRUNC-CVAR | | SOFT-CVAR | | MEAN | |
|---|---|---|---|---|---|---|---|---|---|
| Accuracy | MNIST | **0.99 ± 0.0** | **0.98 ± 0.0** | **0.99 ± 0.0** | **0.98 ± 0.0** | **0.99 ± 0.0** | **0.98 ± 0.0** | **0.99 ± 0.0** | **0.98 ± 0.0** |
| | Fashion-MNIST | **0.99 ± 0.0** | **0.99 ± 0.0** | **0.99 ± 0.0** | 0.98 ± 0.0 | **0.99 ± 0.0** | 0.98 ± 0.0 | **0.99 ± 0.0** | 0.97 ± 0.0 |
| | Cifar-10 | **0.94 ± 0.0** | **0.87 ± 0.0** | 0.59 ± 0.0 | 0.50 ± 0.0 | 0.86 ± 0.1 | 0.64 ± 0.2 | **0.94 ± 0.0** | **0.87 ± 0.0** |
| Surrogate | MNIST | **0.31 ± 0.0** | **0.03 ± 0.0** | 0.35 ± 0.1 | **0.03 ± 0.0** | 0.41 ± 0.1 | 0.04 ± 0.0 | 0.33 ± 0.0 | **0.03 ± 0.0** |
| | Fashion-MNIST | **0.31 ± 0.0** | **0.03 ± 0.0** | 0.67 ± 0.0 | 0.14 ± 0.0 | 0.35 ± 0.0 | 0.04 ± 0.0 | 0.38 ± 0.0 | 0.04 ± 0.0 |
| | Cifar-10 | 2.36 ± 0.1 | **0.25 ± 0.1** | **2.02 ± 0.0** | 1.32 ± 0.0 | 2.79 ± 0.1 | 0.32 ± 0.0 | 2.49 ± 0.1 | **0.24 ± 0.0** |

## E.1 Comparison to other Techniques that Address Distribution Shift

In section 5.2, how ADA-CVAR and the other CVaR optimization algorithms guarded against distribution shift. Now, we compare how do this algorithms compare with algorithms that address distribution shift. We compare against up-sampling and down-sampling techniques for distribution shift for *all* the benchmarks considered in the experiment section. Namely, we perform up-sampling coupled with ADA-CVAR, TRUNC-CVAR, SOFT-CVAR, and MEAN. We expect up-sampling techniques to perform better than raw CVaR optimization when the distribution shift is random *and* the test distribution is *balanced*.

The CVaR guards against *worst-case* distribution shifts for all distributions in the *DRO* set. Certainly, balanced test-sets are contained in the *DRO*, but the *worst-case* guarantees that CVaR brings might be too pessimistic for this particular case. We expect specialized techniques to perform better. Nonetheless, the privileged information that the test-set is balanced might not always be available. In such cases, it could be preferable to optimize for the CVaR of the up-sampled data set, possibly for a large value of $\alpha$.

**Train-Set Shift** In this subsection, we repeat the distribution shift experiment. Here, the train set is shifted (1-to-10 ratio) and the test-set is kept constant. In sake of presentation clarity, we show again the results in table 4. We repeat the experiments but upsampling the training set to balance the dataset. We show the results in table 5. When we compare MEAN with upsampling (last column in table 5) and ADA-CVAR without upsampling (first column in table 6), we see that both methods are comparable. This is because ADA-CVAR addresses this distribution shift algorithmically whereas MEAN does so by re-balancing the dataset. Nevertheless, ADA-CVAR can also re-sample the data set to balance it. This is ADA-CVAR with upsampling (first column in table 5, that outperforms all other algorithms with (and without) upsampling.

Table 4: Test accuracy/loss (mean ± s.d. for 5 random seeds) for train-set distribution shift without train set re-balancing. We highlight the best algorithms.

| | Data Set | ADA-CVAR | | TRUNC-CVAR | | SOFT-CVAR | | MEAN | |
|---|---|---|---|---|---|---|---|---|---|
| Train Set Shift | Adult | **0.65 ± 0.0** | **0.63 ± 0.0** | 0.61 ± 0.1 | 0.69 ± 0.0 | 0.63 ± 0.1 | 0.84 ± 0.3 | 0.60 ± 0.1 | 0.90 ± 0.4 |
| | Australian | **0.68 ± 0.0** | **0.58 ± 0.0** | 0.46 ± 0.1 | 0.70 ± 0.0 | 0.48 ± 0.3 | 0.91 ± 0.4 | 0.46 ± 0.3 | 0.87 ± 0.3 |
| | German | **0.48 ± 0.1** | **0.73 ± 0.0** | 0.44 ± 0.1 | 0.75 ± 0.0 | 0.45 ± 0.1 | 1.00 ± 0.2 | **0.48 ± 0.1** | 0.80 ± 0.1 |
| | Monks | **0.49 ± 0.0** | **0.72 ± 0.0** | 0.39 ± 0.1 | 0.73 ± 0.0 | 0.43 ± 0.3 | 1.27 ± 0.6 | 0.40 ± 0.2 | 0.91 ± 0.3 |
| | Phoneme | **0.58 ± 0.1** | **0.68 ± 0.0** | 0.52 ± 0.2 | 0.69 ± 0.0 | 0.51 ± 0.2 | 1.08 ± 0.4 | 0.55 ± 0.2 | 0.92 ± 0.3 |
| | Spambase | **0.77 ± 0.0** | **0.46 ± 0.0** | 0.74 ± 0.1 | 0.60 ± 0.0 | 0.71 ± 0.2 | 0.61 ± 0.2 | 0.69 ± 0.2 | 0.58 ± 0.2 |
| | Splice | **0.84 ± 0.0** | **0.40 ± 0.0** | 0.77 ± 0.1 | 0.57 ± 0.0 | 0.73 ± 0.1 | 0.52 ± 0.2 | 0.50 ± 0.3 | 0.81 ± 0.4 |
| | Titanic | 0.46 ± 0.2 | **0.70 ± 0.0** | **0.50 ± 0.3** | **0.69 ± 0.0** | 0.43 ± 0.3 | 1.17 ± 0.5 | 0.41 ± 0.3 | 0.87 ± 0.3 |

Table 5: Test accuracy/loss (mean $\pm$ s.d. for 5 random seeds) for train-set distribution shift with train set re-balancing via upsampling We highlight the best algorithms.

| Data Set | ADA-CVAR | | TRUNC-CVAR | | SOFT-CVAR | | MEAN | |
|---|---|---|---|---|---|---|---|---|
| Adult | $0.51 \pm 0.0$ | $\mathbf{0.69 \pm 0.0}$ | $0.62 \pm 0.1$ | $\mathbf{0.69 \pm 0.0}$ | $\mathbf{0.67 \pm 0.1}$ | $0.73 \pm 0.3$ | $\mathbf{0.65 \pm 0.2}$ | $0.78 \pm 0.4$ |
| Australian | $\mathbf{0.77 \pm 0.0}$ | $\mathbf{0.56 \pm 0.0}$ | $0.50 \pm 0.1$ | $0.69 \pm 0.0$ | $0.56 \pm 0.3$ | $\mathbf{0.79 \pm 0.4}$ | $0.54 \pm 0.3$ | $\mathbf{0.77 \pm 0.3}$ |
| German | $\mathbf{0.58 \pm 0.0}$ | $\mathbf{0.68 \pm 0.0}$ | $0.46 \pm 0.1$ | $0.74 \pm 0.0$ | $0.50 \pm 0.2$ | $0.92 \pm 0.2$ | $0.51 \pm 0.1$ | $0.76 \pm 0.1$ |
| Monks | $\mathbf{0.63 \pm 0.0}$ | $\mathbf{0.66 \pm 0.0}$ | $0.42 \pm 0.1$ | $0.72 \pm 0.0$ | $0.49 \pm 0.3$ | $1.10 \pm 0.3$ | $0.45 \pm 0.2$ | $\mathbf{0.85 \pm 0.3}$ |
| Phoneme | $\mathbf{0.65 \pm 0.1}$ | $\mathbf{0.68 \pm 0.0}$ | $0.57 \pm 0.2$ | $\mathbf{0.69 \pm 0.0}$ | $0.57 \pm 0.2$ | $0.94 \pm 0.4$ | $0.60 \pm 0.2$ | $\mathbf{0.82 \pm 0.3}$ |
| Spambase | $\mathbf{0.88 \pm 0.0}$ | $\mathbf{0.43 \pm 0.0}$ | $0.78 \pm 0.1$ | $0.60 \pm 0.0$ | $0.76 \pm 0.2$ | $\mathbf{0.54 \pm 0.2}$ | $0.74 \pm 0.2$ | $\mathbf{0.51 \pm 0.2}$ |
| Splice | $\mathbf{0.89 \pm 0.0}$ | $\mathbf{0.30 \pm 0.0}$ | $0.80 \pm 0.1$ | $0.54 \pm 0.1$ | $0.77 \pm 0.1$ | $0.45 \pm 0.2$ | $0.61 \pm 0.3$ | $0.67 \pm 0.4$ |
| Titanic | $\mathbf{0.51 \pm 0.2}$ | $\mathbf{0.70 \pm 0.0}$ | $\mathbf{0.54 \pm 0.3}$ | $\mathbf{0.69 \pm 0.0}$ | $\mathbf{0.52 \pm 0.3}$ | $1.02 \pm 0.5$ | $\mathbf{0.49 \pm 0.3}$ | $0.80 \pm 0.3$ |

(Row group label: Train Set Shift Upsample)

**Test Shift** Next, we consider another distribution shift. Namely, the train set is kept constant and instead the the test-set is shifted (1-to-10 ratio). In this case, up-sampling should not change the test results. We show the results without re-sampling in table 6 and with upsampling in table 7. In this setting, upsampling does not help because the training sets are balanced. Comparing each case, we see that ADA-CVAR has superior performance to the benchmarks.

Table 6: Test accuracy/loss (mean $\pm$ s.d. for 5 random seeds) for test-set distribution shift without train set re-balancing. We highlight the best algorithms.

| Data Set | ADA-CVAR | | TRUNC-CVAR | | SOFT-CVAR | | MEAN | |
|---|---|---|---|---|---|---|---|---|
| Adult | $0.50 \pm 0.0$ | $\mathbf{0.69 \pm 0.0}$ | $0.64 \pm 0.1$ | $\mathbf{0.69 \pm 0.0}$ | $\mathbf{0.66 \pm 0.1}$ | $\mathbf{0.72 \pm 0.1}$ | $0.64 \pm 0.1$ | $0.77 \pm 0.1$ |
| Australian | $\mathbf{0.76 \pm 0.0}$ | $\mathbf{0.61 \pm 0.0}$ | $0.53 \pm 0.1$ | $0.69 \pm 0.0$ | $0.61 \pm 0.3$ | $0.71 \pm 0.2$ | $0.58 \pm 0.3$ | $0.72 \pm 0.1$ |
| German | $\mathbf{0.52 \pm 0.1}$ | $\mathbf{0.69 \pm 0.0}$ | $0.46 \pm 0.1$ | $0.73 \pm 0.0$ | $0.51 \pm 0.1$ | $0.90 \pm 0.2$ | $\mathbf{0.52 \pm 0.1}$ | $0.75 \pm 0.1$ |
| Monks | $\mathbf{0.63 \pm 0.1}$ | $\mathbf{0.66 \pm 0.0}$ | $0.44 \pm 0.2$ | $0.71 \pm 0.0$ | $0.50 \pm 0.2$ | $0.99 \pm 0.0$ | $0.48 \pm 0.2$ | $0.81 \pm 0.2$ |
| Phoneme | $0.50 \pm 0.2$ | $\mathbf{0.69 \pm 0.0}$ | $0.54 \pm 0.2$ | $\mathbf{0.69 \pm 0.0}$ | $0.56 \pm 0.2$ | $0.92 \pm 0.4$ | $\mathbf{0.58 \pm 0.1}$ | $0.82 \pm 0.3$ |
| Spambase | $\mathbf{0.91 \pm 0.0}$ | $\mathbf{0.52 \pm 0.0}$ | $0.81 \pm 0.1$ | $0.62 \pm 0.0$ | $0.79 \pm 0.1$ | $\mathbf{0.48 \pm 0.2}$ | $0.77 \pm 0.2$ | $\mathbf{0.46 \pm 0.2}$ |
| Splice | $\mathbf{0.91 \pm 0.0}$ | $\mathbf{0.29 \pm 0.0}$ | $0.82 \pm 0.1$ | $0.54 \pm 0.1$ | $0.81 \pm 0.1$ | $\mathbf{0.39 \pm 0.2}$ | $0.67 \pm 0.3$ | $0.58 \pm 0.2$ |
| Titanic | $\mathbf{0.47 \pm 0.2}$ | $\mathbf{0.70 \pm 0.0}$ | $\mathbf{0.52 \pm 0.3}$ | $\mathbf{0.69 \pm 0.0}$ | $\mathbf{0.52 \pm 0.3}$ | $0.99 \pm 0.2$ | $\mathbf{0.50 \pm 0.3}$ | $0.81 \pm 0.2$ |

(Row group label: Test Set Shift)

Table 7: Test accuracy/loss (mean $\pm$ s.d. for 5 random seeds) for test-set distribution shift with train set re-balancing via upsampling. We highlight the best algorithms.

| Data Set | ADA-CVAR | | TRUNC-CVAR | | SOFT-CVAR | | MEAN | |
|---|---|---|---|---|---|---|---|---|
| Adult | $0.46 \pm 0.0$ | $\mathbf{0.69 \pm 0.0}$ | $\mathbf{0.65 \pm 0.1}$ | $\mathbf{0.69 \pm 0.0}$ | $\mathbf{0.66 \pm 0.1}$ | $\mathbf{0.72 \pm 0.1}$ | $0.64 \pm 0.1$ | $0.76 \pm 0.1$ |
| Australian | $\mathbf{0.76 \pm 0.0}$ | $\mathbf{0.61 \pm 0.0}$ | $0.55 \pm 0.1$ | $0.68 \pm 0.0$ | $0.64 \pm 0.3$ | $0.67 \pm 0.4$ | $0.61 \pm 0.3$ | $0.69 \pm 0.3$ |
| German | $\mathbf{0.52 \pm 0.1}$ | $\mathbf{0.69 \pm 0.0}$ | $0.46 \pm 0.1$ | $0.73 \pm 0.0$ | $\mathbf{0.52 \pm 0.1}$ | $0.89 \pm 0.2$ | $\mathbf{0.53 \pm 0.1}$ | $0.75 \pm 0.1$ |
| Monks | $\mathbf{0.63 \pm 0.1}$ | $\mathbf{0.66 \pm 0.0}$ | $0.46 \pm 0.2$ | $0.71 \pm 0.0$ | $0.51 \pm 0.2$ | $0.92 \pm 0.5$ | $0.49 \pm 0.2$ | $0.79 \pm 0.2$ |
| Phoneme | $0.50 \pm 0.2$ | $\mathbf{0.69 \pm 0.0}$ | $0.52 \pm 0.2$ | $\mathbf{0.69 \pm 0.0}$ | $0.55 \pm 0.2$ | $0.90 \pm 0.4$ | $\mathbf{0.57 \pm 0.1}$ | $0.82 \pm 0.3$ |
| Spambase | $\mathbf{0.91 \pm 0.0}$ | $\mathbf{0.54 \pm 0.0}$ | $0.82 \pm 0.1$ | $0.63 \pm 0.0$ | $0.80 \pm 0.1$ | $\mathbf{0.44 \pm 0.2}$ | $0.79 \pm 0.2$ | $\mathbf{0.43 \pm 0.2}$ |
| Splice | $\mathbf{0.92 \pm 0.0}$ | $\mathbf{0.28 \pm 0.0}$ | $0.84 \pm 0.1$ | $0.53 \pm 0.1$ | $0.83 \pm 0.1$ | $\mathbf{0.36 \pm 0.2}$ | $0.71 \pm 0.3$ | $0.53 \pm 0.4$ |
| Titanic | $\mathbf{0.47 \pm 0.2}$ | $\mathbf{0.70 \pm 0.0}$ | $\mathbf{0.52 \pm 0.3}$ | $\mathbf{0.69 \pm 0.0}$ | $\mathbf{0.52 \pm 0.3}$ | $0.99 \pm 0.2$ | $\mathbf{0.50 \pm 0.3}$ | $0.81 \pm 0.2$ |

(Row group label: Test Set Shift Upsample)

**Double Shift** Finally, we consider a case where the train set is imbalanced (1-to-10 ratio), and the test set is even more imbalanced (1-to-100 ratio). We show the results without re-sampling in table 8 and with upsampling in table 9. In this case, we upsampling is detrimental. ADA-CVAR clearly outperforms all the other algorithms with upsampling and without upsampling.

Table 8: Test accuracy/loss (mean $\pm$ s.d. for 5 random seeds) for imbalanced train and test-sets without train set re-balancing. We highlight the best algorithms.

| Data Set | ADA-CVAR | | TRUNC-CVAR | | SOFT-CVAR | | MEAN | |
|---|---|---|---|---|---|---|---|---|
| Adult | **0.94 ± 0.0** | **0.50 ± 0.0** | 0.70 ± 0.2 | 0.69 ± 0.0 | 0.70 ± 0.2 | 0.63 ± 0.3 | 0.69 ± 0.2 | 0.66 ± 0.4 |
| Australian | **0.90 ± 0.0** | **0.39 ± 0.0** | 0.59 ± 0.2 | 0.67 ± 0.1 | 0.69 ± 0.3 | 0.59 ± 0.4 | 0.66 ± 0.3 | 0.61 ± 0.3 |
| German | **0.95 ± 0.0** | **0.43 ± 0.0** | 0.52 ± 0.2 | 0.70 ± 0.1 | 0.58 ± 0.2 | 0.79 ± 0.3 | 0.59 ± 0.2 | 0.69 ± 0.2 |
| Monks | **0.95 ± 0.0** | **0.48 ± 0.1** | 0.50 ± 0.2 | 0.70 ± 0.0 | 0.58 ± 0.3 | 0.81 ± 0.4 | 0.56 ± 0.3 | 0.72 ± 0.3 |
| Phoneme | **0.58 ± 0.3** | **0.67 ± 0.0** | **0.57 ± 0.2** | 0.69 ± 0.0 | **0.61 ± 0.2** | 0.79 ± 0.4 | **0.63 ± 0.2** | 0.73 ± 0.3 |
| Spambase | **0.98 ± 0.0** | **0.19 ± 0.0** | 0.85 ± 0.1 | 0.60 ± 0.1 | 0.83 ± 0.1 | 0.39 ± 0.2 | 0.82 ± 0.2 | 0.39 ± 0.2 |
| Splice | **0.98 ± 0.0** | **0.18 ± 0.0** | 0.86 ± 0.1 | 0.52 ± 0.1 | 0.85 ± 0.1 | 0.31 ± 0.2 | 0.75 ± 0.3 | 0.46 ± 0.4 |
| Titanic | **0.85 ± 0.2** | **0.49 ± 0.0** | 0.52 ± 0.3 | 0.69 ± 0.0 | 0.59 ± 0.3 | 0.85 ± 0.3 | 0.57 ± 0.3 | 0.74 ± 0.3 |

(rows grouped under: Double Set Shift)

Table 9: Test accuracy/loss (mean $\pm$ s.d. for 5 random seeds) for imbalanced train and test-sets with train set re-balancing via up-sampling. We highlight the best algorithms.

| Data Set | ADA-CVAR | | TRUNC-CVAR | | SOFT-CVAR | | MEAN | |
|---|---|---|---|---|---|---|---|---|
| Adult | 0.40 ± 0.1 | **0.69 ± 0.0** | **0.68 ± 0.2** | **0.69 ± 0.0** | **0.72 ± 0.2** | **0.60 ± 0.3** | **0.71 ± 0.2** | **0.62 ± 0.4** |
| Australian | **0.86 ± 0.0** | **0.46 ± 0.0** | 0.61 ± 0.2 | 0.66 ± 0.1 | 0.71 ± 0.3 | **0.56 ± 0.2** | 0.67 ± 0.3 | 0.59 ± 0.2 |
| German | **0.78 ± 0.1** | **0.58 ± 0.0** | 0.55 ± 0.2 | 0.69 ± 0.1 | 0.60 ± 0.2 | 0.75 ± 0.2 | 0.60 ± 0.2 | 0.68 ± 0.1 |
| Monks | **0.68 ± 0.1** | **0.64 ± 0.0** | 0.51 ± 0.2 | 0.69 ± 0.0 | 0.59 ± 0.2 | **0.78 ± 0.3** | 0.57 ± 0.2 | **0.72 ± 0.2** |
| Phoneme | 0.49 ± 0.3 | **0.69 ± 0.0** | **0.59 ± 0.2** | **0.69 ± 0.0** | **0.63 ± 0.2** | 0.75 ± 0.2 | **0.65 ± 0.2** | **0.70 ± 0.2** |
| Spambase | **0.95 ± 0.0** | **0.26 ± 0.1** | 0.86 ± 0.1 | 0.60 ± 0.1 | 0.84 ± 0.1 | 0.36 ± 0.2 | 0.83 ± 0.2 | **0.37 ± 0.2** |
| Splice | **0.96 ± 0.0** | **0.18 ± 0.0** | 0.87 ± 0.1 | 0.50 ± 0.1 | 0.86 ± 0.1 | **0.29 ± 0.2** | 0.77 ± 0.3 | 0.43 ± 0.2 |
| Titanic | **0.62 ± 0.3** | **0.68 ± 0.0** | **0.53 ± 0.3** | **0.69 ± 0.0** | **0.58 ± 0.3** | 0.82 ± 0.4 | **0.57 ± 0.3** | **0.72 ± 0.3** |

(rows grouped under: Double Set Shift Upsample)

**Experiment Conclusion** Techniques that address class imbalance, such as up-sampling, are useful when there is a-priori knowledge that the test-set is balanced (cf. Train Shift experiment). When this is not the case, such techniques may be detrimental (cf. Double Shift experiment). ADA-CVAR, and the CVaR DRO optimization problem, is orthogonal to techniques for imbalanced dataset and can be used together, as shown by the previous experiments. Overall, ADA-CVAR also has lower standard errors than MEAN and SOFT-CVAR. This indicates that it is more robust to the random sampling procedures.

# F More Related Work

## F.1 Combinatorial Bandit Algorithms

A central contribution of our work is an *efficient* sampling algorithm based on an instance of combinatorial bandits, the $k$-set problem. In this setting, the learner must choose a subset of $k$ out of $N$ experts with maximum rewards, and there are $\binom{N}{k}$ such sets. Koolen et al. (2010) introduce this problem in the full information setting ($k$-sets) and Cesa-Bianchi and Lugosi (2012) extend it to the bandit setting. Cesa-Bianchi and Lugosi (2012) propose an algorithm, called CombBand, that attains a regret of $O(k^{3/2}\sqrt{NT\log(N/k)})$ when the learner receives bandit feedback. Audibert et al. (2013) prove a lower bound of $O(k\sqrt{NT})$, which is attained by Alatur et al. (2020) up to a $\sqrt{\log(N)}$ factor. However, the computational and space complexity of the CombBand algorithm is $O(kN^3)$ and $O(N^3)$, respectively. Uchiya et al. (2010) propose an efficient sampling algorithm that has $O(N\log(k))$ computational and $O(N)$ space complexity. Instead, we efficiently represent the algorithm proposed by Alatur et al. (2020) using Determinantal Point Processes (Kulesza et al., 2012). This yields an algorithm with $O(\log(N))$ computational and $O(N)$ space complexity.

## F.2 Sampling from $k$DPPs

We propose to directly sample from the marginals of the (relaxed) $k$-DPP. Our setting has the advantage that the k-DPP is diagonal and there is no need for performing an eigendecomposition of the kernel matrix, which takes $O(N^3)$ operations. However, there are efficient methods that avoid the eigendecomposition and return a sample (of size $k$) of the $k$-DPP. Sampling uniformly at random from such sample can be done in constant time. State-of-the-art exact sampling methods for k-DPPs take at least $O(N \operatorname{poly}(k))$ operations using rejection sampling (Dereziński et al., 2019), whereas approximate methods that use MCMC have mixing times of $O(Nk)$ (Anari et al., 2016). Hence, such algorithms are inefficient compared to our sampling algorithm that takes only $O(\log(N))$. This is because our method is specialized on diagonal $k$-DPPs and latter algorithms remain valid for general $k$-DPPs.