[Reviews · NeurIPS 2020]

Review 1

Summary and Contributions: This paper proposes an aadaptive sampling algorithm for optimizing CVaR of a loss function. The sampler uses EXP3 on top of k-DDP to maximize the loss.

Strengths: The framework can be practical in robust/fair learning.

Weaknesses: The algorithm isn't as elegant as one would hope. Well-known existing methods are stacked to get expected results. Sampling only one data point in one iteration seems impractical. Is it possible to use a minibatch?

Correctness: The mathematical claims of the paper seems correct.

Clarity: Yes.

Relation to Prior Work: This has a flavor similar to boosting/GAN in that the learner is playing a game against sampler which forces the learner to be more robust. Is there any connection to AdaBoost? How about GANs? Combinatorial multi-armed bandit literature?

Reproducibility: Yes

Additional Feedback: Did you consider combinatorial bandit algorithms instead of k-DPP + EXP3 ? How many repeat runs did you do? Figure 3 has really large error bars. Looking at the loss distributions of AdaCVaR vs Mean would be a more interesting visualization than the test score bars, both for Figure 2 and Figure 3. Figure 2 "Classification" results are hard to interpret. Does that mean AdaCVaR prefers uniform-like predictions on easy data?


Review 2

Summary and Contributions: The paper proposes an adaptive sampling algorithm to optimize the conditional value-at-risk, a risk measure that focuses on the elements with the highest loss. The approach proposes an alternative sampling method, that bypass the high variance problem of the standard batch-based estimators of the conditional value-at-risk. The theoretical guarantees are usual generalization bounds or average regret bounds. The approach is supported by numerical experiments.

Strengths: The paper addresses an important problem from the perspective of a minimax game between a learner and a sampler. It is well-written and accessible. I believe that the claim is new, and the empirical performance evaluation is done with other papers on CVaR optimization. ---- After rebuttal ----- I read ofter reviews, the author's response slightly modified my score.

Weaknesses: Since the paper features both ideas from statistical learning and bandit algorithms, its understanding requires theoretical background in both fields.

Correctness: The claims and method, as well as the empirical methodology, are correct.

Clarity: The paper is well-written, but there are some complicated or undefined definitions. Precisely: - l.95 \hat{C} is not defined before it is presented. - l.120 could you give a reference for the 'partial bandit feedback model' ? - Lots of notation abuse, e.g. \mathbb{E}_q - In Algorithm 1, \hat{L}_t is probably the estimator of L_t in R^N, but it is assigned a real value, - In Algorithm 1, what is \sim_{u.a.r.} ? - l. 160 defines W_{I, t} while l. 167 defines W_{t, I} which is quite confusing. Typos: - l 264: 'Hence it finds' - avoid inline fractions (\frac{}{})

Relation to Prior Work: The relation to prior work is well-explained.

Reproducibility: Yes

Additional Feedback:


Review 3

Summary and Contributions: ## Update ## Thank you very much for kindly responding to my comments. In my opinion, the proposed method itself is interesting and well analyzed. On the other hand, the method is not motivated enough in light of the DRO context: How the proposed approach is advantageous over existing methods with other DRO sets? Is the theoretical result stronger than those of existing methods? All in all, I feel the paper is "marginally" above, and I will maintain my score. ########### This paper proposes an adaptive sampling method that enables us to stochastically optimize CVaR of loss function values, which involves two challenges: the rejection of many sampled data points and exploding gradients. The underlying idea for avoiding them is to write the problem in a min-max form and to use no-regret algorithms. A naive implementation of a sampler's online algorithm, which samples data points, incurs an exponential computation cost. The authors avoid this problem with sampling from a k-DPP kernel. The proposed method is validated with experiments.

Strengths: - The resulting CVaR value is theoretically bounded in expectation. - The empirical effectiveness is confirmed with extensive experiments. - Risk-averse learning is an important topic relevant to the NeurIPS community.

Weaknesses: - In the context of robust optimization, the idea of using min-max formulations and no-regret algorithms is prevalent. In this sense, the proposed method appears to be just a CVaR variant of the line of work. Thus it is somewhat weak in terms of technical novelty and significance. - As in Section 2, Namkoong & Duchi (2016) proposed a stochastic DRO method for DRO sets with Cressie-Read f-divergence, and it is stated that the DRO set considered in this work is different. However, it is unclear why employing the different DRO set is advantageous.

Correctness: The claims and methods seem to be correct, although I did not check the proofs in Appendix.

Clarity: The paper is mostly well-written and the statement is clear. The second paragraph in Section 1 is a little bit misleading, where the authors raised "non-convexity" and "high variance" as current issues. As far as I can see, no theoretical results that resolve the issues are presented in the paper.

Relation to Prior Work: The relation to prior work is well described, but the authors should elaborate more on differences from other DRO methods (e.g., Namkoong & Duchi (2016)). Comparisons with other combinatorial bandit algorithms (e.g., [Combes et al., Combinatorial Bandits Revisited, NIPS2015]) should also be presented to explain why the k-DPP-based method is advantageous.

Reproducibility: Yes

Additional Feedback: Is it possible to obtain any theoretical guarantee that bounds the variance of CVaR (and its gradient)?

[Author Response · NeurIPS 2020]

We would like to thank all reviewers for their valuable feedback and comments. Please find our responses below.

**Reviewer 1** - Use of mini-batches: in our experiments, we indeed use mini-batches of size $B$, by sampling $B$ points independently according to $q$.

- Comparison to AdaBoost: we use the general strategy to solve minimax games used in the AdaBoost paper, as we explain in line 119. Note that AdaBoost trains and combines multiple models (weak learners), whereas we train a single model via stochastic optimization, with points sampled adaptively. AdaBoost optimizes the empirical risk, whereas we minimize the CVaR.

- Comparison to Combinatorial Bandits: we indeed use a particular combinatorial bandit algorithm for $m$-sets as discussed and cited in lines 153-155. We also provide a detailed comparison to other combinatorial bandit approaches in appendix F.1.

- Relation to GANs: Both approaches solve minimax games (as do many other ML approaches), but there is no deeper connection.

- Algorithm elegance: we find the algorithm quite simple and efficient as it is basically SGD with an adaptive sampling distribution which is efficiently updated.

- Number of runs: As explained in Appendix D, we do runs over 5 different random seeds.

- Large error bars: These are only in the trunc-cvar and soft-cvar algorithms, which arise from their high variance (which is one of their disadvantages).

- Interpretation of results in Fig. 2: The CVaR of Trunk-CVaR is the lowest, but the predictive accuracy is very low. This is because it predicts an almost uniform distribution. AdaCVaR instead has slightly worse CVaR, but obtains a very good predictive accuracy. AdaCVaR also has a lower CVaR than ERM (standard SGD). For detailed results please refer to Table 1 in Appendix E.

**Reviewer 2** - Definition of $\hat{C}$. Thank you for observing that. It is expressed in (3), but not spelled out.

- Reference for the 'partial bandit feedback model': In Freund and Shapire 1999 they introduce such model. But maybe Lattimore and Szepesvári 2018 is a more modern introduction. We will include it when we introduce the model.

- $\hat{L}_t$ is in $\mathbb{R}^N$ and is all zeros expect in the index $i = i_t$. We will clarify the notation $[[i = i_t]]$.

- $\sim_{u.a.r.}$ means sampled uniformly at random.

- $W_{I,t}$ vs $W_{t,I}$: Yes you are correct, we will fix these nomenclature issues.

**Reviewer 3** - Difference with Namkoong & Duchi (2016): The problem we address is different. Our goal is to efficiently solve the CVaR optimization problem due to its wide use as a criterion in many applications. To do so, we use a DRO formulation. The goal in N&D (2016) is to ensure the robustness of ERM w.r.t. a family of distributions. To do so they use a DR-formulation of ERM. Notice that the DR family of distributions considered by N&D (2016) *do not* contain the CVaR. Importantly, this family of distributions induces a convex structure whereas the CVaR induces a combinatorial structure, hence the algorithm we use is also different. We will clarify this difference when discussing the related work.

- Non-Convexity: We do not solve the problem of non-convexity. We propose an approach that reduces the CVaR problem to a (sequence of) Empirical Risk Minimization (ERM) problems on weighted data. Thus, as long as the resulting non-convex ERM problems can be solved, we are able to solve the (non-convex) CVaR. Empirically, SGD finds good solutions for non-convex ERM problems that arise in deep learning, hence such reduction is useful.

- High-variance: As discussed in line 120 and lines 291-299, the trunc-cvar has gradients with higher magnitude, thus stochastic gradients have higher variance. To address this, we propose adaptive sampling to reduce the variance of such gradients (avoiding the multiplication by 1/alpha). Such importance-sampling strategies are common to reduce variance. In our case, the importance sampling is adaptive because the model is changing through the optimization process.

- Other combinatorial bandits: We have a discussion in Appendix F.1. We will elaborate on the relationship with Combes et al., Combinatorial Bandits Revisited (2015) in the revised version. In particular, the work by Combes et al. considers general combinatorial sets and the regret incurred for the particular case of $m$-sets is higher than our algorithm by an extra $m^{1/2}$ term, which is important for $m = \alpha N$. Furthermore, the algorithm requires to compute a pseudo-inverse of a $N \times N$ matrix at each iteration which is prohibitive for large scale applications and it invalidates the benefits of stochastic optimization.

[Meta-Review · NeurIPS 2020]

This paper is quite on the borderline and could go either way. All the 3 reviewers and the meta reviewer deem reach a consistent agreement that this paper is borderline. The meta reviewer suggests an accept but would not mind if the chairs decide the other way.